# TCSIF: A temporally consistent global GOME-2A solar-induced chlorophyll fluorescence dataset with correction of sensor degradation

Chu Zou[1,2,3], Shanshan Du[1,2], Xinjie Liu[1,2], Liangyun Liu[1,2,3]

[1] Key Laboratory of Digital Earth Science, Aerospace Information Research Institute, Chinese Academy of Sciences, Beijing,100094, China

[2] University of Chinese Academy of Sciences, Beijing 100049, China

[3] International Research Center of Big Data for Sustainable Development Goals, Beijing 100094, China

*Correspondence to: Xinjie Liu (liuxj@radi.ac.cn)*

**Abstract.** Satellite-based solar-induced chlorophyll fluorescence (SIF) serves as a valuable proxy for monitoring the photosynthesis of vegetation globally. Global Ozone Monitoring Experiment-2A (GOME-2A) SIF product has gained widespread popularity, particularly due to its extensive global coverage since 2007. However, serious temporal degradation of the GOME-2A instrument is a problem, and for now, there is a lack of time-consistent GOME-2A SIF products that meet the needs of temporal trend analysis. In this paper, the GOME-2A instrument's temporal degradation was first calibrated using a pseudo-invariant method, which revealed 16.21 % degradation of the GOME-2A radiance at the near-infrared (NIR) band from 2007 to 2021. Based on the calibration results, the temporal degradation of the GOME-2A radiance spectra was successfully corrected by using a fitted quadratic polynomial function whose determination coefficient ($R^2$) is 0.851. Next, a data-driven algorithm was applied for SIF retrieval at the 735–758 nm window. Besides, a photosynthetically active radiation (PAR)-based upscaling model was employed to upscale the instantaneous clear-sky observations to monthly average values to compensate for the changes in cloud conditions and atmospheric scattering. Accordingly, a global GOME-2A SIF dataset (TCSIF) with correction of temporal degradation was successfully generated from 2007 to 2021, and the spatiotemporal pattern of global SIF was then investigated. Corresponding trend maps of the global temporally consistent GOME-2A SIF showed that 62.91 % of vegetated regions underwent an increase in SIF, and the global annual averaged SIF exhibited a trend of increasing by 0.70 % $yr^{-1}$ during the 2007–2021 period. The TCSIF dataset is available at https://doi.org/10.5281/zenodo.8242928 (Zou et al., 2023).

**Keywords:** GOME-2 satellite; temporal degradation; trend analysis; Solar-induced chlorophyll fluorescence

## 1 Introduction

Solar-induced chlorophyll fluorescence (SIF) retrieved from satellite-based hyperspectral data provides a new way to proxy the photosynthesis of vegetation globally. Numerous studies have demonstrated that satellite-based SIF observations are able to produce better estimates of gross primary productivity (GPP) than the widely used reflectance-based approaches (Sun et al., 2017; Guanter et al., 2014; Zhang et al., 2014).

Currently, the satellite sensors used for SIF retrieval can be generally divided into two types according to their spectral resolution (Frankenberg et al., 2011; Frankenberg et al., 2014; Guanter et al., 2012; Du et al., 2018). The first type of satellite was originally designed to measure the atmospheric XCO2 concentration using observations with a spectral resolution higher than 0.05 nm; these satellites include GOSAT (Frankenberg et al., 2011; Guanter et al., 2012), OCO-2 (Frankenberg et al., 2014; Sun et al., 2017), TanSat (Du et al., 2018), and OCO-3 (Taylor et al., 2020). The other type of satellite instrument was originally designed for atmospheric chemistry applications and had a spectral resolution of about 0.5 nm. These instruments

included the Global Ozone Monitoring Experiment 2 (GOME-2) onboard the MetOp-A/B/C satellites (Joiner et al., 2013; Joiner et al., 2016; Köhler et al., 2015); the SCanning Imaging Absorption spectroMeter for Atmospheric CHartograhY (SCIAMACHY) onboard the ENVIronmental SATellite (ENVISAT) (Köhler et al., 2015; Joiner et al., 2016); and the TROPO-spheric Monitoring Instrument (TROPOMI) onboard the Sentinel-5P satellite (Köhler et al., 2018). Given that its global coverage capability starts from 2007, the GOME-2 satellite-based SIF dataset has been the most widely used for global monitoring of GPP, crop yield, drought, vegetation phenology, etc. (Sun et al., 2015; Guanter et al., 2014; Yoshida et al., 2015; Lu et al., 2018; Chen et al., 2019). Yet, the volatile coating used within GOME-2's optical bench enclosure makes the optical lens more susceptible to contamination, which eventually leads to instrument degradation (A. Hahne; Munro et al., 2016). Further, such degradation may affect the solar and Earth radiance measurements in different ways, depending on the optical components involved, and correcting this via the onboard calibration method may be impossible (Munro et al., 2016). Moreover, how degradation impacts the quality of different level-2 products is highly dependent on the individual algorithms used. Generally, there is a strong decreasing trend in the GOME-2A level-2 SIF product as derived from the GOME-2A level-1B radiance product. For example, the GOME-2A SIF generated by Joiner et al. (2016) as well as the Sun-Induced Fluorescence of Terrestrial Ecosystems Retrieval (SIFTER) SIF dataset produced by Sanders et al. (2016) were both found to harbor an artificial trend caused by instrument degradation (Zhang et al., 2018; Koren et al., 2018). For example, Yang et al. (2018) reported the SIF emission of the Amazon forests decreased during the 2015/2016 El Niño event when analyzed using the GOME-2 SIF data by Joiner et al. (2016), which is in conflict with the increase of the enhanced vegetation index (EVI) and downward solar shortwave radiation. Zhang et al. (2018) argued that the reduced GOME-2A SIF signal in the Amazon Forest observed by Yang et al. (2018) could have been caused by artifacts associated with the temporal degradation of the GOME-2A instrument, instead of an actual decline in photosynthesis. Hence, it is imperative to address the temporal-decreasing artifact of the GOME-2A dataset before its application to any analysis and interpretation of interannual trends.

Researchers have tried to generate consistent long-term SIF datasets. For example, Wang et al. (2022) assembled a long-term consistent global SIF dataset (LT_SIFc*) by combining the global SIF products from GOME, SCIAMACHY, and GOME-2. The temporal degradation problem was corrected based on the satellite SIF measurements over the Sahara Desert between 1995 and 2018. Unfortunately, this attempt is not sufficiently rigorous, in that the degradation of sensors does not transit to SIF in a linear manner due to post-processing processes. Furthermore, the LT_SIFc* is a reprocessed product derived from existing GOME-2 SIF products, which limits its temporal resolution to 1 month and hinders its broader application. Earlier, Schaik et al. (2020) applied a seasonal factor to GOME-2 reflectance and retrieved SIF from that temporally-corrected reflectance data to generate the SIFTER v2 product; however, the function fitted with the season as the smallest unit may entail deviations from the actual reality of sensor degradation. Accordingly, in terms of the processing results, significant interannual variation persists in the SIFTER v2 time series (Wang et al., 2022). Presently, we still lack a robust consistent long-term GOME-2 SIF product that has been generated via rigorous recalibration methods and can yield reasonable, meaningful results. This leaves the long-term observations provided by GOME-2 underutilized scientifically.

The objective of this study was to provide a temporally consistent GOME-2A SIF dataset that overcomes the degradation problem, spanning 2007 to 2021. Temporal degradation of GOME-2A level-1B radiance was first calibrated using the pseudo-invariant method in the Sahara Desert. Then a data-driven approach was applied to retrieve the SIF datasets from the corrected GOME-2A measurements. Finally, a global temporally consistent monthly GOME-2A SIF (TCSIF) dataset for 2007–2021 was generated, using the PAR-based temporal upscaling method, from the degradation-corrected GOME-2A instantaneous SIF retrievals. The temporally consistent GOME-2A SIF dataset generated here offers a promising tool for monitoring global vegetation variation from 2007 through 2021 and it will advance our understanding of vegetation's photosynthetic activities at a global scale.

## 2 Datasets

### 2.1 Datasets for the generation of TCSIF

GOME-2A (launched on October 19th, 2006) was designed by the European Space Agency to measure atmospheric ozone, trace gases, and ultraviolet radiation. Since 2007, it has been collecting top-of-atmosphere (TOA) radiance data spanning a spectral range of 270 to 790 nm from four channels (Munro et al., 2006). Of these, channel 4 of GOME-2A has a spectral coverage of 593–790 nm wavelengths with a spectral resolution of 0.48 nm, which was successfully used to generate a global SIF dataset (Joiner et al., 2013).

The MODIS Version 6.1 Nadir Bidirectional reflectance distribution Adjusted Reflectance (NBAR) product (MCD43C4) (Schaaf et al., 2002) records the surface reflectance at a nadir viewing angle for each pixel at local solar noon. It has a spatial resolution of 0.05° × 0.05° and a daily temporal resolution (Schaaf et al., 2002). The MODIS NBAR product is considered stable over long periods of time and was used here to investigate the homogeneity and stability of the calibration site (see Sect. 3.1).

The EVI product derived from the MODIS Vegetation Indices 16-Day (MOD13C1) Version 6.1 with a spatial resolution of 0.05° was aggregated to 0.5° (Didan, 2021). PAR was obtained from the Merra-2 meteorological assimilation reanalysis data (Gelaro et al., 2017) and this PAR dataset had a spatial resolution of 0.5° × 0.625° (resampled to 0.5° × 0.5°) and a temporal interval of 1 h. The EVI product and Merra-2 PAR dataset were used to upscale the instantaneous SIF to monthly values, as described in Sect. 3.4.

### 2.2. Datasets for evaluation and comparison

The dataset was verified through a two-step verification, i.e., the verification of the corrected radiance (compared to radiance measurements in the absence of sensor degradation) and SIF retrievals (compared to other long-term products).

Radiance spectra obtained from GOME-2C serve as a benchmark for the calibrated GOME-2A radiance. Being a sensor that measures the same bands with the same spectral resolution as GOME-2A, GOME-2C has a later launching time in November 2018. Thus, measurements at the initial launch stage of GOME-2C can be taken as accurate values that are not affected by degradation.

The NDVI (normalized difference vegetation index) and three global GPP products were utilized for validation purposes. We employed the global NDVI derived from the MOD13C1 product. The MOD17A2H GPP (MODIS GPP) product, with a spatial resolution of 500 m (Running et al., 2021), was mosaicked globally every 8 days during the 2007–2021 period. Global-simulated GPP based on the LUE model (Pmodel GPP) is a daily product from 1982 to 2016, whose spatial resolution is 0.5° (Stocker et al., 2019). The monthly, 0.5° GPP derived from the Dynamic Global Vegetation Model (DGVM) for 2007 to 2021 was also utilized (TRENDY GPP Version 11) (Sitch et al., 2015). The temporal range, temporal resolution, and spatial resolution of these datasets are summarized in Table 1. All these products were resampled at a spatial resolution of 0.5° and a temporal resolution of 1 month to enable their comparison.

**Table 1 GPP and NDVI datasets used in this study and their relevant details.**

| Dataset | Temporal range | Temporal resolution | Spatial resolution |
|---------|----------------|---------------------|--------------------|
| MODIS GPP | 2000.2–2023.2 | 8 days | 500 m |
| Pmodel GPP | 1982.1–2016.12 | 1 day | 0.5° |
| TRENDY GPP | 1900.1–2021.12 | 1 month | 0.5° |
| MODIS NDVI | 2000.2–2023.2 | 16 days | 0.05° |

Next, we selected four long-term SIF products spanning more than one decade for comparison, including the LT_SIFc*(1995–2018) (Wang et al., 2022), SIFTER v2 (2007–2018) (Schaik et al., 2020), GOSIF (2000–2022) (Li and

Xiao, 2019), and GOME-2 SIF products generated by the National Aeronautics and Space Administration (hereon abbreviated as NASA SIF) (2007–2018) (Joiner et al., 2013&2016). The LT_SIFc* is a data fusion product of GOME, SCIAMACHY, and GOME-2, having a spatial resolution of 0.05° and a temporal resolution of 1 month. It dealt with the temporal decay of the instrument based on statistics of SIF signals in the Sahara Desert. The SIFTER v2 product is the point-by-point SIF product retrieved from GOME-2 measurements after applying a time-related correction factor; it was composited to yield a 0.5°, monthly global map in this study. The GOSIF product is the spatiotemporal extrapolation product based on the global neural network model and OCO-2 SIF V8r product, with a spatial resolution of 0.05° and a temporal resolution of 8 days. Apart from the SIF products spanning decades, the OCO-2 SIF product from 2015 to 2021, and TROPOMI SIF from 2018 to 2021 are also included here for comparative purposed given its high accuracy and it is being less affected by sensor degradation. All SIF products were resampled to a 0.5°, monthly spatiotemporal resolution and were compared with TCSIF to assess long-term trends in this study. Additionally, we used the NASA GOME-2A level 2 SIF product, which has not been corrected for temporal decay, to verify the spatial distribution of our product. Key information about these SIF products is presented in Table 2.

**Table 2 SIF products used in this study and their relevant details.** This information includes the temporal range of the dataset; whether the dataset initially had a temporal degradation problem, and if so, whether the degraded dataset was corrected. The signal to which the correction factor is directly applied, the temporal unit of the correction factor, and the function describing the temporal correction are provided as well.

| Dataset | Temporal range | Temporal degradation problem? | Temporal correction applied? | Signal directly corrected | Temporal unit | Function |
|---|---|---|---|---|---|---|
| TCSIF | 2007.1–2021.11 | Yes | Yes | Radiance | 1 day | Quadratic function |
| NASASIF | 2007.1–2019.3 | Yes | No | - | - | - |
| LT_SIFc* | 1995.1–2018.12 | Yes | Yes | SIF | 1 month | Ensemble Empirical Mode Decomposition approach |
| SIFTER | 2007.1–2018.12 | Yes | Yes | Reflectance | 3 months | Piecewise function |
| GOSIF | 2000.3–2022.12 | No | - | - | - | - |
| OCO-2 SIF | 2014.9–2021.12 | No | - | - | - | - |
| TROPOMI SIF | 2018.4–2022.12 | No | - | - | - | - |

## 3. Methods

### 3.1 Pseudo-invariant method for calibrating the GOME-2A degradation

A homogeneous square region in the Sahara Desert (22.5°–23.5° E, 28.5°–29.5° N; Figure 1b) was selected as a pseudo-invariant site for calibrating the GOME-2A degradation. Ignoring the spatiotemporal variation in the far-red surface reflectance and atmospheric optical properties over the calibration site during the 2007–2021 period, the temporal trend of TOA GOME-2A reflectance could be deemed equivalent to the amount of temporal degradation in the GOME-2A

instrument.

The MCD43C4 product was used here to investigate the homogeneity and stability of this calibration site. Figure 1b depicts the MCD43C4 surface reflectance and its spatiotemporal variance for the calibration site in 2007–2021. These results indicate this site is bright (the near-infrared [NIR] reflectance is high, at 55.3 %–60.6 %), homogeneous (with mean spatial variation = 0.29 %), and stable (with very low temporal variation = 0.81 %). Arguably, this site qualified as an ideal calibration site for implementing the pseudo-invariant method.

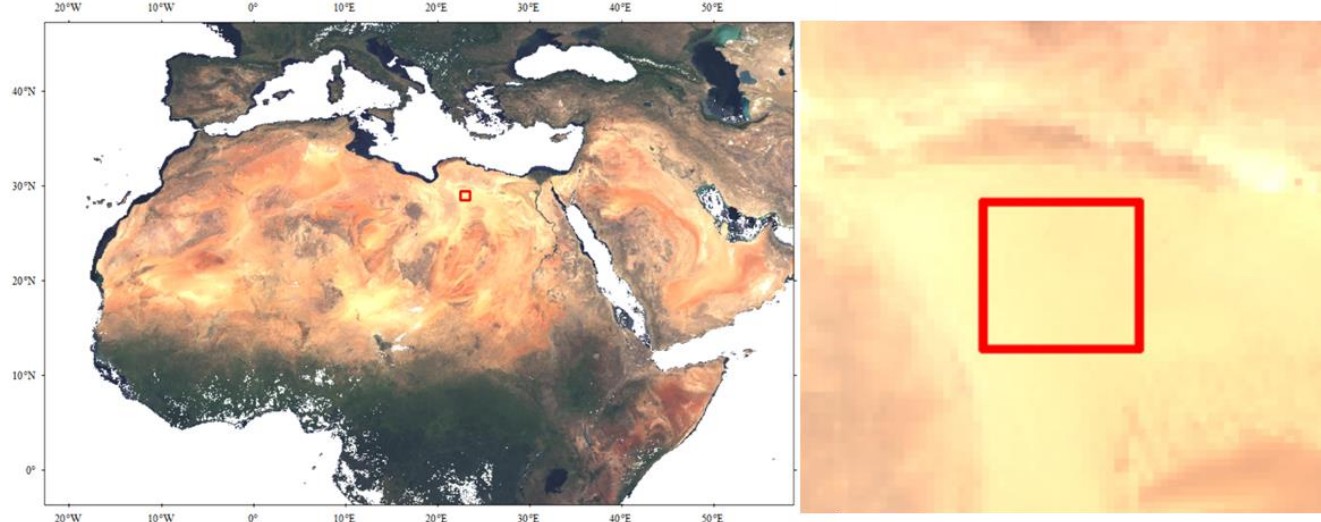

(a) Location of the calibration site in the Sahara Desert

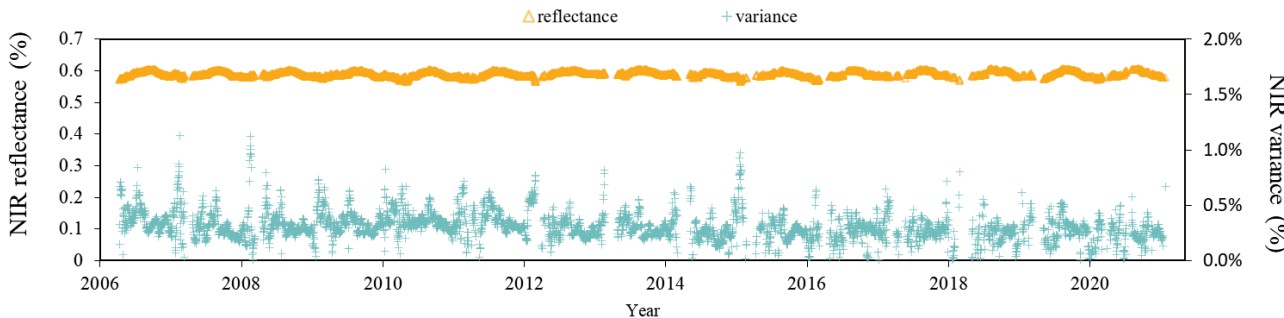

(b) Temporal variation in NIR reflectance at the calibration site

**Figure 1.   (a) Location of the calibration sites. (b)The NIR surface reflectance and its temporal variance at the calibration site (22.5°–23.5° E, 28.5°–29.5° N) during the 2007–2021 period. The NIR reflectance (shown by yellow triangles) and the NIR variance (shown by blue crosses) are respectively the mean and variance of surface reflectance at the near-infrared band.**

The clear-sky GOME-2A level-1B radiance products for the calibration site during 2007–2021 were downloaded to derive the temporal degradation. Two selection criteria for the GOME-2A data were applied: (1) a scanning angle $< 20°$, and (2) no cloud contamination. This resulted in a total of 6885 GOME-2A level-1B radiance spectra being collected to correct for the GOME-2A degradation.

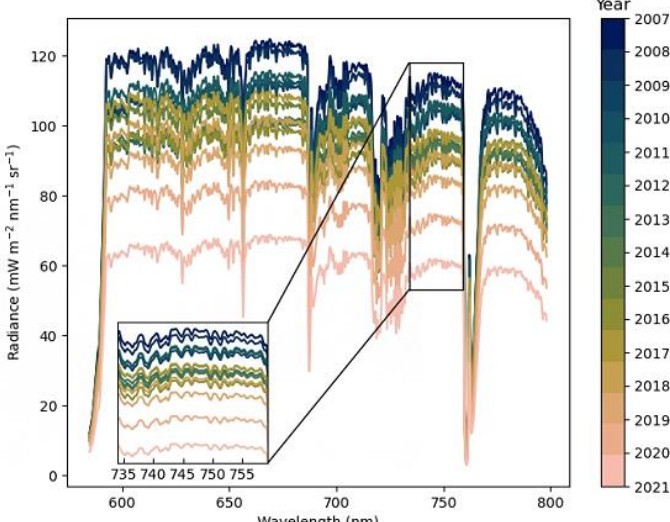

**Figure 2. Temporal variation in the GOME-2A level-1B top-of-atmosphere (TOA) radiance spectra at the calibration site (22.5°–23.5° E, 28.5°–29.5° N) for the 2007–2021 period. Different colors represent different years from 2007 to 2021.**

Figure 2 depicts the yearly averaged TOA radiance spectra over the calibration site for each year in 2007–2021. Temporal degradation was determined using GOME-2A level-1B radiance products in the near-infrared (NIR) band between 735 and 758 nm, which served as the fitting window for SIF retrieval. Evidently, there is pronounced temporal degradation in the radiance spectra. Thus, a time-dependent correction factor was calculated, and the temporal correction function was assumed to be a second-order polynomial as follows:

$$\text{Dfactor} = a \cdot NOD^2 + b \cdot NOD + c, \tag{1}$$

where Dfactor is the degradation correction factor describing the temporal degradation. NOD is the number of elapsed days since January 1st, 1900, starting with 1. a, b, and c are the fitting coefficients of the polynomial function based on the near-infrared radiance of the pseudo-invariant site, the detailed analysis can be found in Sect.4.1.

Next, the GOME-2A radiance can be corrected by dividing the measured radiance signal by the Dfactor:

$$Rad_c(\text{NOD}, \lambda) = \frac{Rad_o(\text{NOD}, \lambda)}{\text{Dfactor}(\text{NOD})}, \tag{2}$$

where $Rad_c$ and $Rad_o$ are respectively the corrected radiance and original radiance without correction for the degradation; Dfactor is the degradation correction factor in Equation (1), used to compensate for the GOME-2A instrument's degradation since 2007.

## 3.2 Data-driven based SIF retrieval method

The TCSIF dataset was separated from far-red SIF and corrected radiance spectra in the 735–758 nm range by using an SVD-based data-driven approach, namely that proposed by Guanter et al. (2015).

The TOA radiance ($L_{TOA}$) was modeled this way:

$$L_{TOA} = \left( \sum_{i=0}^{n_p} a_i \cdot \lambda^i \right) \cdot \left( \sum_{j=1}^{n_{pc}} \beta_j \cdot v_j \right) + F_s \cdot h_F \cdot T_\uparrow^e, \tag{3}$$

where $L_{TOA}$ is the TOA radiance at 735–758 nm; $\lambda$ is the measured wavelength used to represent the low-frequency information in surface reflectance and atmospheric scattering; and $v_j$ is the $j$-th singular vector derived from non-vegetated targets (referred to as training datasets) describing the high-frequency information in solar irradiance and atmospheric transmittance. The $\alpha_i$ and $\beta_j$ are the coefficients of the polynomial and singular vectors, respectively; $F_s$ is the SIF intensity at 740 nm; $\lambda$ is the wavelength; $n_p$ is the order of the polynomial; and $n_{pc}$ is the number of singular vectors selected. Finally,

$T_{\uparrow}^e$ is the effective upward transmittance estimated as follows (Köhler et al., 2015):

$$T_{\uparrow}^e = exp\left[ln\left(T_{\downarrow\uparrow}^e \cdot \frac{\sec(\theta_v)}{\sec(\theta_0) + \sec(\theta_v)}\right)\right],$$

(4)

where $T_{\downarrow\uparrow}^e$ is the effective two-way atmospheric transmittance derived by normalizing the TOA reflectance using the low-order polynomial function; $\theta_0$ and $\theta_v$ denote the solar zenith angle and viewing zenith angle, respectively.

### 3.3 Post-processing of SIF retrieval results

The following quality-filtering criteria were applied (Guanter et al., 2012):

(1) Land cover type is set to vegetation;

(2) Range of the mean radiance within the 735–758nm window is between 25 and 200 mW m$^{-2}$ nm$^{-1}$ sr$^{-1}$;

(3) Absolute value of SIF is <5 mW m$^{-2}$ nm$^{-1}$ sr$^{-1}$;

(4) Solar zenith angle is <75°;

(5) $\chi^2$ is <2.

Here, $\chi^2$ is the reduced chi-square value calculated based on the residuals of fitting (Sun et al., 2018), which characterizes the fit between the modeled and measured radiance using the forward model described above, in Equation (3). Its calculation is given by:

$$\chi^2 = \frac{\sum_i^{n_{wi}}(\frac{(Rad_{fit}^i - Rad_{true}^i)^2}{noise})^2}{n_f},$$

(5)

where $Rad_{fit}^i$ and $Rad_{true}^i$ denote the $i$-th spectral point of the modeled and measured radiance within the fitting window, respectively; $noise$ denotes the random noise spectra; $n_f$ is the degrees of freedom, and $n_{wi}$ is the number of bands within the fitting window.

Besides, we dealt with the effect of a zero-offset error in the SIF retrievals. The spectrometer radiance signals' nonlinear response and the SVD data-driven algorithm can inevitably introduce systematic biases to SIF retrieval results, especially so in non-vegetated areas. Previous studies have identified systematic biases in SIF retrievals that depend on either the TOA radiance (Frankenberg et al., 2011; Guanter et al., 2012; Sun et al., 2017; Sun et al., 2018) or latitude (Köhler et al., 2015; Joiner et al., 2016; Schaik et al., 2020). Here we corrected the systematic biases (bias) by considering the radiance at the 735–758 nm window (Rad), latitude (lat), and observation zenith angle ($\theta_0$) of each footprint as follows (Joiner et al., 2016):

$$\frac{bias}{\cos(\theta_0)} = A + B \cdot \theta_0 + C \cdot \theta_0^2 + D \cdot \theta_0^3 + E \cdot Rad + F \cdot Rad^2 + G \cdot Rad^3 + H \cdot lat,$$

(6)

where A to H are the correction factors. These factors were firstly determined using the training dataset (where SIF is supposed to be zero and the retrieved SIF can be taken as "bias") which are uniform in latitude dimension by applying the least squares model. Next, the bias was calculated and subtracted from SIF retrievals for each pixel.

### 3.4 Accuracy evaluation of the product

First, the root mean square of the model residual (RMS_residual) was used to assess the accuracy of the data-driven model to fit the radiance spectra. The model residual (Res) is the difference between the modeled and measured radiance:

$$Res(\lambda) = Rad_{true}(\lambda) - Rad_{fit}(\lambda),$$

(7)

where $Rad_{fit}^i$ and $Rad_{true}^i$ denote the modeled and measured radiance spectra, respectively.

Second, the covariance matrix $S_e$ of the least squares for SIF retrieval was calculated to assess the precision of SIF retrievals:

$$S_e = noise^2(K^\mathrm{T}K)^{-1}, \tag{8}$$

where $K$ is the Jacobian matrix formed by those linear model parameters from Eq. (3), and *noise* refers to the spectrally uncorrelated noise, which was calculated here based on the radiance and signal-to-noise ratio.

The standard error of the weighted mean ($\sigma_{\mathrm{SIF}}$) within each grid cell was calculated this way (Du et al., 2018):

$$\sigma_{\mathrm{SIF}} = \frac{1}{\sqrt{\sum_1^n (1/\sigma_i^2)}}, \tag{9}$$

where $\sigma_i$ is the 1–σ error, which is the diagonal element of $S_e$ corresponding to F$_\mathrm{s}$, and $n$ is the number of sample points within each grid cell.

## 3.5 Upscaling the instantaneous SIF to the monthly averaged value

In previous studies, the global satellite-observed SIF was upscaled to a daily scale by using the diurnal cycle of the cosine of the solar zenith angle (cos[SZA]) to correct for day-length effects (Frankenberg et al., 2011; Zhang et al., 2018). These effects can cause large overestimates of SIF on cloudy days because the satellite-observed SIF data are only available on clear-sky days. In this study, the downwelling PAR rather than cos(SZA) was used to compensate for the significant effects of diurnal weather changes due to cloud and atmospheric scattering (Hu et al., 2018) while upscaling the instantaneous SIF to monthly values. The all-sky monthly averaged SIF (SIF$_{\mathrm{mon}}$) can be determined using the PAR-based upscaling model, as follows:

$$\mathrm{SIF}_{\mathrm{mon}} = \begin{cases} \frac{\sum_{\mathrm{mon}}^M \mathrm{SIF}_{\mathrm{ins}}}{\sum_{\mathrm{mon}}^M \mathrm{PAR}_{\mathrm{ins}} \times \mathrm{EVI}_{\mathrm{ins}}} \times \mathrm{PAR}_{\mathrm{mon}} \times \mathrm{EVI}_{\mathrm{mon}}, \text{if } \mathrm{EVI}_{\mathrm{mon}} > 0.2 \\ \\ \frac{\sum_{\mathrm{mon}}^M \mathrm{SIF}_{\mathrm{ins}}}{\sum_{\mathrm{mon}}^M \mathrm{PAR}_{\mathrm{ins}}} \times \mathrm{PAR}_{\mathrm{mon}}, \text{if } \mathrm{EVI}_{\mathrm{mon}} \leq 0.2 \end{cases}, \tag{10}$$

where SIF$_{\mathrm{ins}}$ is the GOME-2A level-2 daily instantaneous clear-sky (i.e., <30 % cloud fraction) SIF; the terms PAR$_{\mathrm{mon}}$ and PAR$_{\mathrm{ins}}$ are the corresponding monthly and instantaneous values of PAR; and EVI$_{\mathrm{mon}}$ and EVI$_{\mathrm{ins}}$ are the respective monthly and daily EVI values. $M$ is the number of valid measurements within the 0.5° grid cell during the relevant monthly period. The EVI is negligible if the EVI value for the cell is <0.2.

Based on the PAR-based upscaled model, the instantaneous GOME-2A SIF clear-sky observations with a correction of temporal degradation were upscaled to their monthly average values.

## 4 Results

### 4.1 Correction of GOME-2A sensor degradation

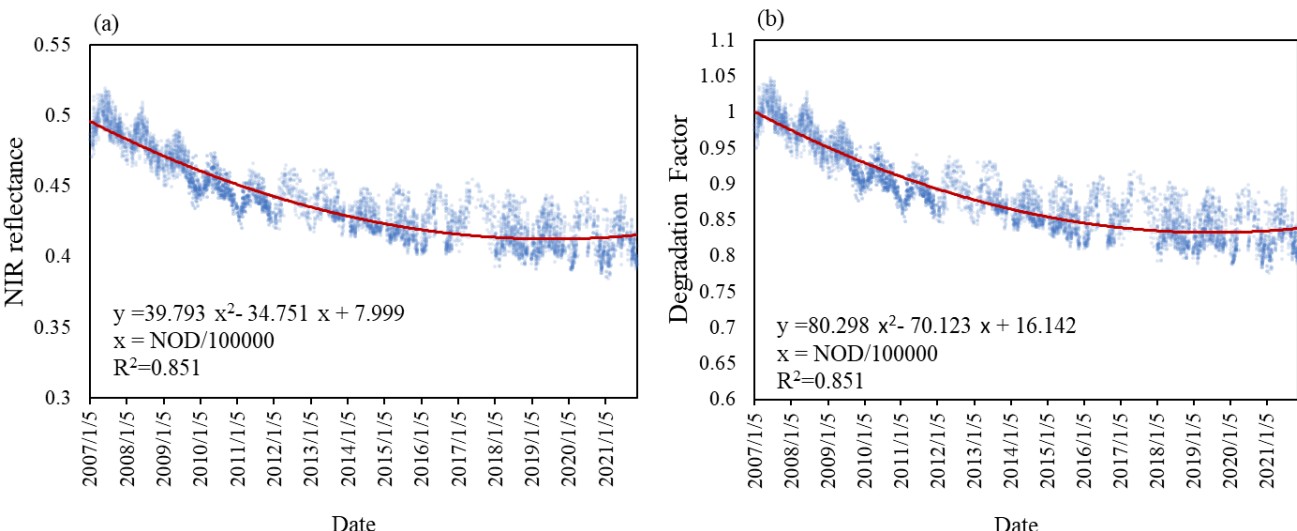

**Figure 3. Temporal variation in the TOA reflectance at 758 nm at the calibration site (left) and the temporal correction coefficients used to compensate for the degradation of GOME-2A since 2007 (right). The blue dots and red curves in (a) represent NIR reflectance and the fitted quadratic function. The degradation factor (Dfactor) in (b) were calculated by dividing the NIR reflectance by the value of the quadratic function in (a) at the starting date (1 January 2007). "NOD" in the degradation correction equation is the GOME-2A acquisition date since 2007, which equals the number of days from 1 January 1900.**

A second-order polynomial was fitted to describe the temporal degradation in the reflectance signal of GOME-2A. Figure 3a illustrates the temporal variation in TOA reflectance at 758 nm at the calibration site. Significant and continuous degradation can be observed; however, this nonlinear trend could be accurately captured by a quadratic polynomial function with a determination coefficient ($R^2$) of 0.851. These results indicated that, overall, the GOME-2A instrument degraded by 16.21 % from 2007 to 2021. This temporal degradation was considered spectrally constant in the narrow fitting window of SIF retrieval (735–758 nm).

By dividing the NIR reflectance by the value of the fitted function at the starting date (1 January 2007), we obtain the degradation factor (Dfactor), as shown in Figure 3b. The second-order polynomial fitted was used in Eq. (2) to calibrate the instrument's degradation since 1 January 2007, as given by:

$$\text{Dfactor(NOD)} = 80.298 \times \left(\frac{\text{NOD}}{100000}\right)^2 - 70.123 \times \frac{\text{NOD}}{100000} + 16.142, \ (R^2 = 0.851), \tag{11}$$

where "NOD" is the number of days since January 1st, 1900.

The temporally corrected GOME-2A NIR radiance was validated using GOME-2C radiance spectra (Figure 4). For the corrected GOME-2A radiance, the scatter plot shows that the majority of points are concentrated near the 1:1 line (Figure 4a). The difference between the two products followed a Gaussian distribution with a small mean value of 1.85 mW m$^{-2}$ sr$^{-1}$ nm$^{-1}$, which is 2.3% of the mean GOME-2A radiance (Figure 4b). On the contrary, the mean deviation without temporal correction is 15.16W m$^{-2}$ sr$^{-1}$ nm$^{-1}$ (Figure 4d). Slight positive offsets can be found in both linear regression results. The difference in orbit height between GOME-2A (827 km) and GOME-2C (817 km) leads to the difference in viewing zenith angle (VZA). Although only observations with VZA<20° were selected, and the effect of observing angle has been corrected by dividing the cosine of VZA, there may still be differences due to the anisotropy of the ground surface, which introduces systematic errors.

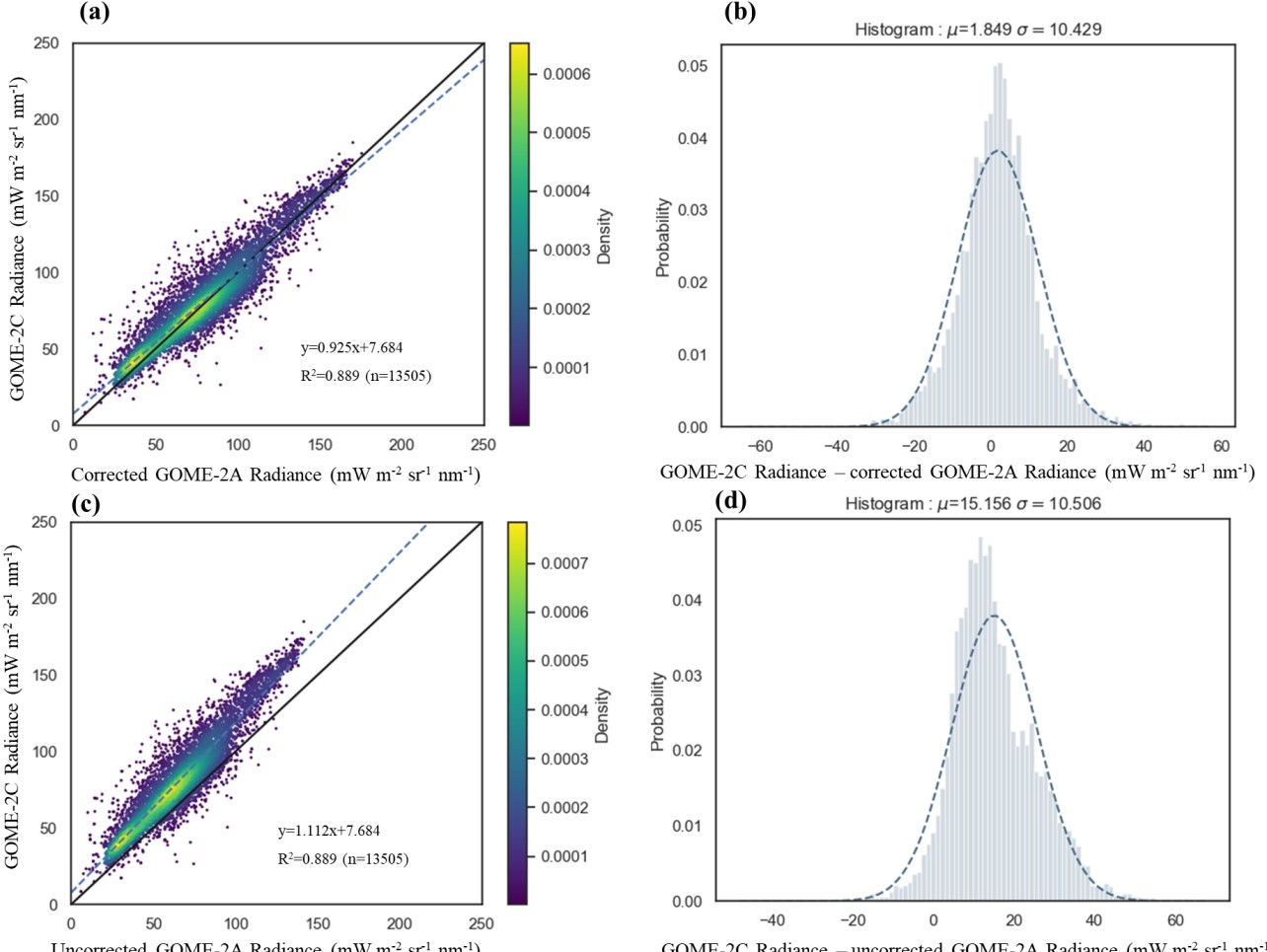

**Figure 4 Comparison between GOME-2A and GOME-2C NIR radiance after (a,b) and before (c,d) the temporal correction on 1 July 2019. The histogram of difference for GOME-2C NIR radiance minus the corrected and uncorrected GOME-2A NIR radiance was shown in (b) and (d), respectively. Spatially matched pixels with cloud fraction of lower than 0.3, VZA of lower than 20° and SZA of lower than 70° were selected.**

## 4.2 Uncertainty of the data-driven algorithm

The fitting residual and single retrieval error of the TCSIF dataset was analyzed to verify the feasibility of the data-driven retrieval algorithm as well as the quality control process.

As Figure 5 shows, the fitted data-driven model described well the measured radiance spectra, with a root mean square (RMS) of the residual that was below 0.30 %. The model considering fluorescence is better capable of reconstructing the radiance spectra than that ignoring fluorescence, with a slightly lower RMS_residual (around 0.02 % on average).

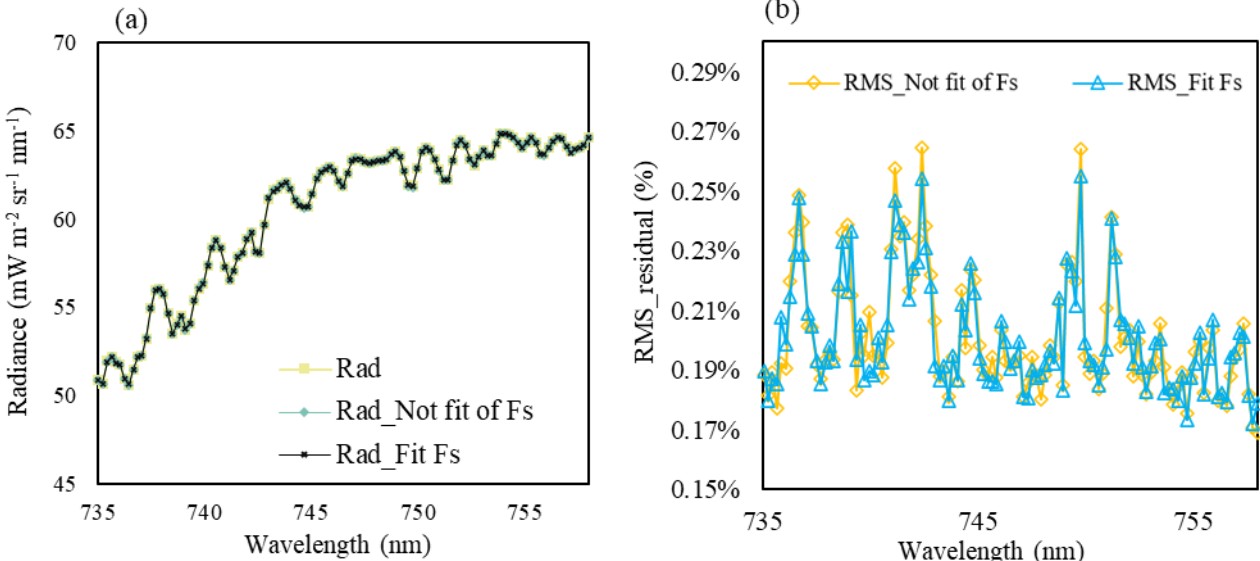

**Figure 5. (a) Measured (Rad, represented by square marks) and modeled (with [Rad_Fit Fs, represented by damond marks] and without [Rad_Not Fit of Fs, represented by crosses] accounting for SIF) radiance spectra in the 735–758 nm fitting window over vegetated areas on 15 July 2017. (b) Root mean square (RMS) of the fitting residual with (RMS _Fit Fs, represented by yellow diamonds) and without (RMS_Not Fit of Fs, represented by blue triangles) accounting for SIF. The spectra are the average of 224 vegetation spectra over pixels with a cloud fraction < 0.1 and SIF intensity > 1.5 mW m⁻² sr⁻¹ nm⁻¹.**

### 4.3 Spatial distribution of the TCSIF dataset

Figure 6 shows the global pattern of monthly TCSIF in the summer and winter of 2008. The monthly GOME-2A SIF dataset captured well the spatial patterning in both seasons, in which Southeast Asia, the North American Corn Belt, and Central Europe in July, and the Amazon Rainforest and most of South America in December, all showed high SIF values. Crucially, the standard error of the weighted mean ($\sigma(F_s)$) is lower than 0.1 mW m$^{-2}$ sr$^{-1}$ nm$^{-1}$ in most regions globally, while the main vegetated areas have $\sigma(F_s)$ of lower than 0.05 mW m$^{-2}$ sr$^{-1}$ nm$^{-1}$ (Figure 6).

We also compared the spatially matched TCSIF and NASA SIF pixels in January and July 2008, July 2017, and January 2018 (Figure 7a–d). The linear relationships between the two SIF products revealed these to be strongly correlated ($R^2 > 0.65$), significant ($p$-value < 0.05), and close to the 1:1 correspondence line (slope > 0.84) for either season in 2008 (Figure 7a, b). For comparison in 2017 and 2018 (Figure 7c, d), there are still good linear relationships between the TCSIF and NASA SIF ($R^2 > 0.64$). However, it is worth noticing that the regression line deviates from the 1:1 line in both 2017 and 2018 (slope < 0.80), which was caused by the degradation in NASA SIF.

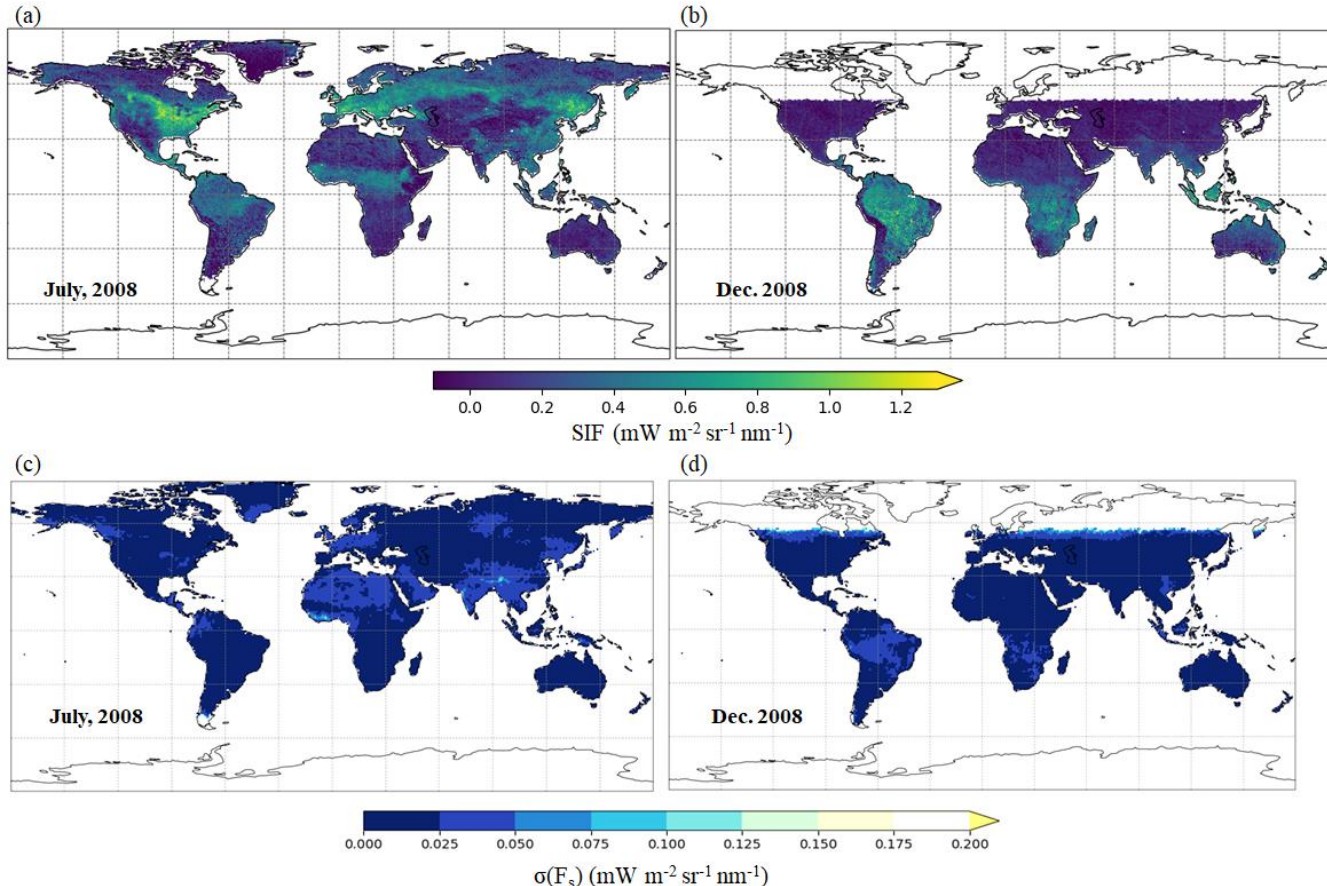

**Figure 6. Global patterns in the upscaled monthly TCSIF (a, b) and standard error of the weighted mean ($\sigma(F_s)$) (c, d) in July (a, c) and December (b, d) in the year 2008.**

OCO-2 SIF and TROPOMI SIF were also involved in the validation of TCSIF (Figure 7e,f). To avoid discrepancies in wavelength and the overpassing time, the day-length corrected 740 nm provided by OCO-2 SIF, TROPOMI SIF, and TCSIF were compared. The spatially matched points were selected. TCSIF versus OCO-2 SIF and TCSIF versus TROPOMI SIF comparisons were conducted in July 2019 and July 2021, respectively. Both comparisons show high consistency with $R^2 >$ 0.65, and the linear regression results are close to the 1:1 line.

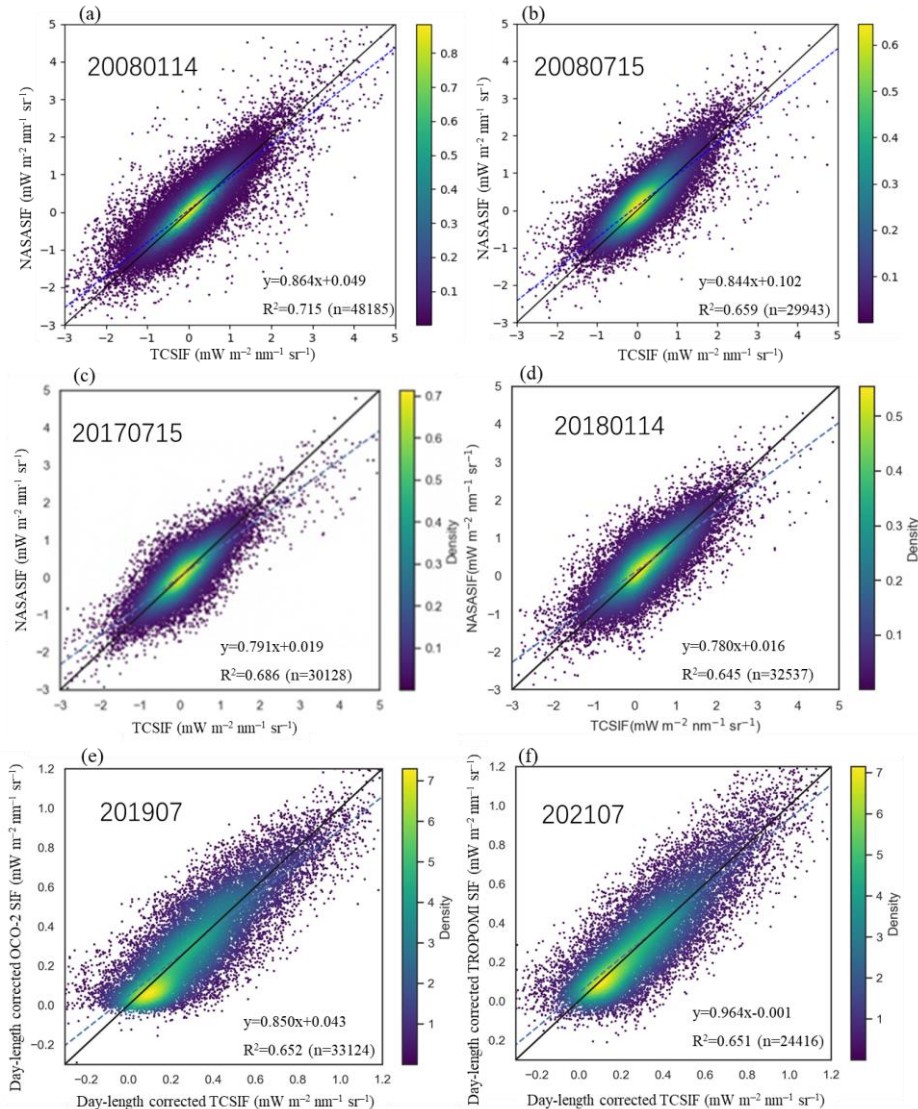

**Figure 7. Comparison of TCSIF vs. NASA SIF on 14 January (a) and 15 July (b) in the year 2008, 15 July 2017 (c), and 14 January 2018 (d). Comparison of TCSIF versus OCO-2 in July 2019 (e) and TCSIF versus TROPOMI SIF in July 2021 (f). The comparison was made based on the level 2 product. Co-located pixels over land with a cloud fraction < 0.3 have been selected. The color of the scatter points represents the density of the points. The blue dotted line and the black solid line represent the line fitted based on the scatter points and the 1:1 line, respectively.**

## 4.4 Temporal variation in the TCSIF dataset

The global monthly SIF is averaged to demonstrate temporal variation (Figure 8). The autocorrelation coefficient of the time series is calculated for each pixel, and only the vegetation-covered pixels with an autocorrelation coefficient greater than 0.4 are selected to ensure the authenticity of the time series. Compared with the NASASIF products, which gave a downward trend of SIF for 2007–2018, the global monthly mean trend of TCSIF exhibited an upward trend. The monthly trend in global averaged SIF shifted from one of decreasing by 1.15 % yr$^{-1}$ to one of increasing by 0.71 % yr$^{-1}$ after correcting the instrument's degradation. As seen in Figure 9, the trend in SIF's variation in almost all vegetation regions was underestimated before the temporal correction, with the effect of the correction being particularly prominent at low latitudes in the Southern Hemisphere (0°–20° S), as well as at middle and high latitudes in the Northern Hemisphere (30°–70° N).

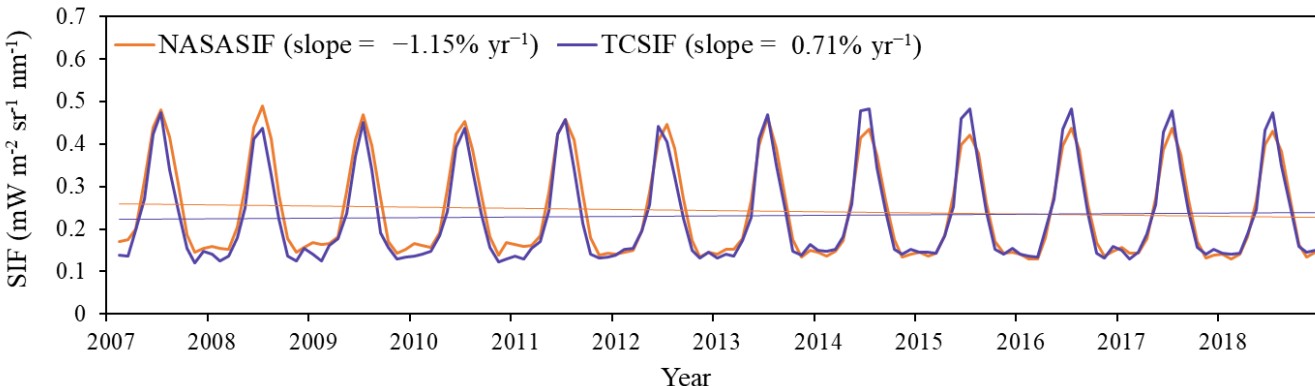

**Figure 8. Time series of the monthly averaged global GOME-2A SIF, with (purple line) and without (orange line) the degradation correction, for 2007–2018. The daily level 2 NASASIF product was composited and filtered in the same way as for TCSIF.**

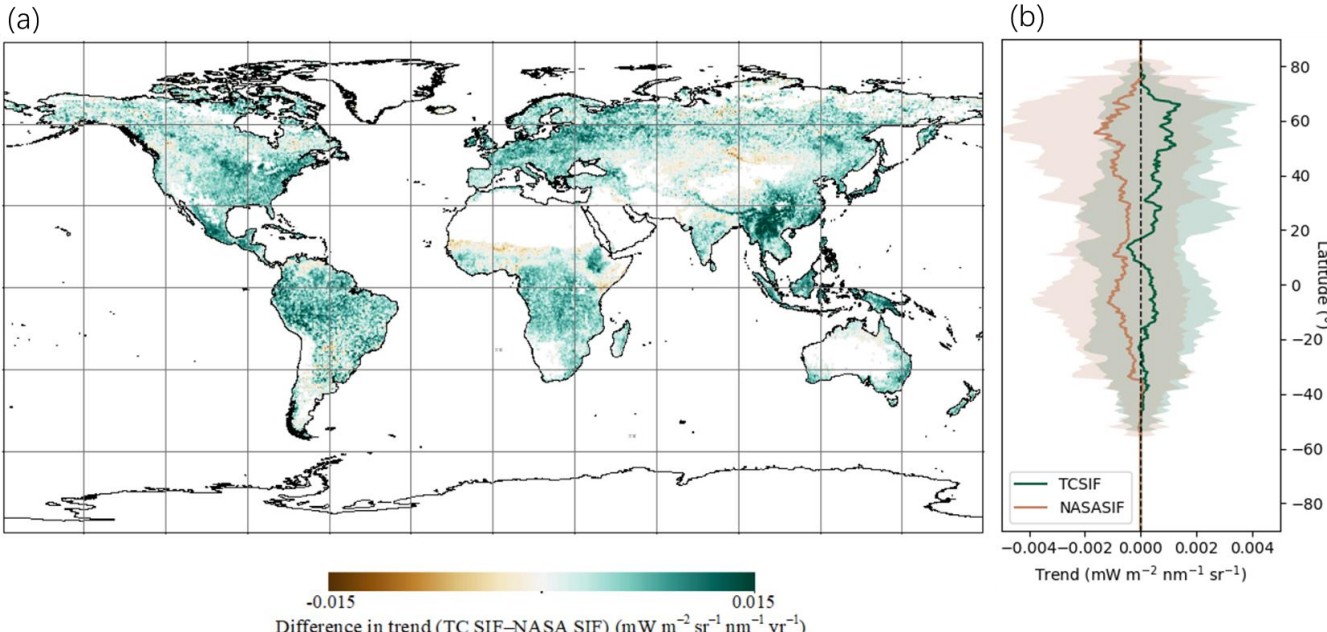

Figure 9. (a) Difference in the temporal trend between SIF products with and without temporal correction, and (b) the latitudinal profiles of (a) for 2007–2018. The brown and green shaded areas in (b) represent the standard deviation of the TCSIF and NASA SIF trends, respectively.

The temporally consistent SIF dataset was then applied to reveal spatiotemporal patterns in the photosynthetic activity of global vegetation. Figure 10 shows the global patterns in the trends for the annual average TCSIF in the 2007–2021 period. When tallied, 62.91 % of the vegetation areas overall were distinguished by an upward trend of SIF, whereas 13.86 % corresponded to a significant increase over time ($p < 0.05$). Those regions undergoing a significant increase in SIF were mainly located in Southeast Asia, Eastern China, Western Europe, Central Africa, and South America. Only 4.51 % of the vegetated parts of the Earth's vegetated surface experienced a significant decrease in SIF.

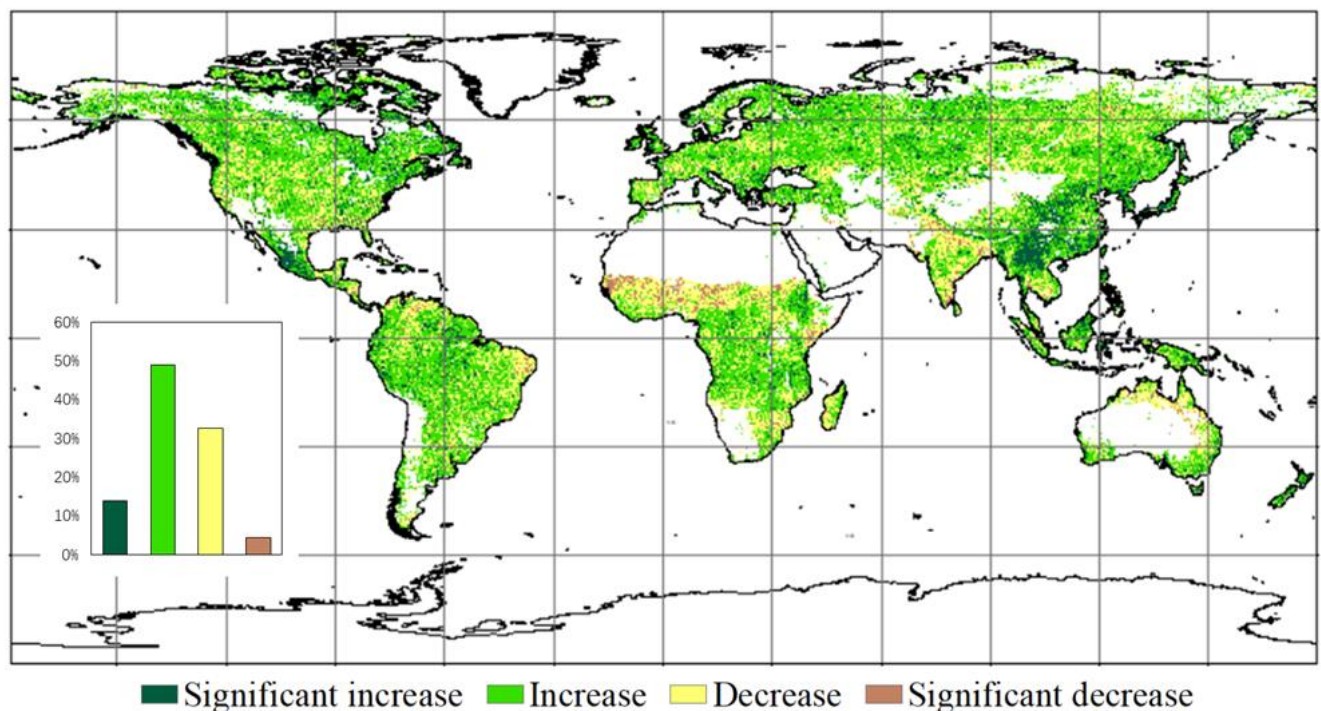

**Figure 10. Map of trends in the annual average GOME-2A SIF for 2007–2021. The inset shows the percentage of areas characterized by four types of trends (significant increase: positive correlation with $p < 0.05$; increase: positive correlation with $p \geq 0.05$; decrease: negative correlation with $p \geq 0.05$; significant decrease: negative correlation with $p < 0.05$).**

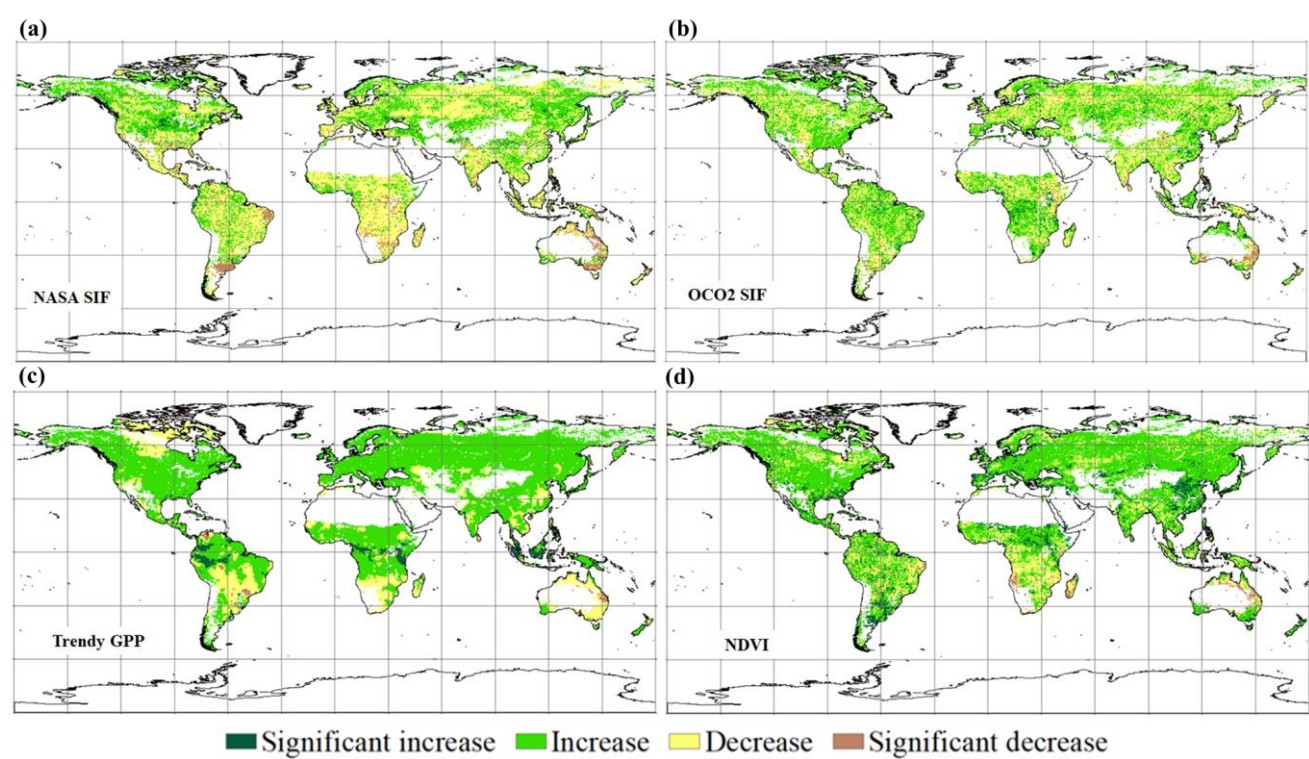

**Significant increase** **Increase** **Decrease** **Significant decrease**

**Figure 11. Map of trends in the annual average (a)NASA SIF for 2007–2018, (b) OCO-2 SIF for 2015–2021, as well as (c) trendy GPP and (d) NDVI for 2007–2021. The colors represent four types of trends (significant increase: positive correlation with $p < 0.05$; increase: positive correlation with $p \geq 0.05$; decrease: negative correlation with $p \geq 0.05$; significant decrease: negative correlation with $p < 0.05$).**

As shown in Figure 11, 57.11% of vegetation areas are facing a decline in NASA SIF. On the opposite, as seen from OCO-2 SIF, trendy GPP, and NDVI, vegetation is growing over a large area globally (>70%) from 2007 (or 2015) to 2021.

The main inconsistency between NASA SIF and the other products occurs in central and southern Africa, eastern Europe, and southern North America, where NASA SIF declines and the others increase. In southeastern China, vegetation greenings were found by TCSIF, OCO-2 SIF, and NDVI, while an insignificant downward trend was shown by trendy GPP. Vegetation growth in southern North America, Europe, the Amazon rainforest, central Africa, and Southeast Asia was detected by all the products apart from NASA SIF.

## 5 Discussion

### 5.1 Degradation at different locations and wavelengths

In this study, only one calibration site (Libya 4) was used for the fitting of the degradation function. The results may be different for different sites. Previous studies have compiled 20 pseudo-invariant calibration sites(PICS) for instrument calibration (Cosnefroy et al., 1996; Bacour et al., 2019). We have involved three other commonly used sites, and the related information is shown in Table 3.

**Table 3 Four pseudo-invariant calibration sites(PICS) and related information**

| Site name | location | NIR reflectance | mean spatial variation | temporal variation |
|---|---|---|---|---|
| Libya 4 | (23.00° E, 29.00° N) | 55.3 %–60.6 % | 0.29% | 0.81% |
| Algeria 3 | (7.66° E, 30.33° N) | 49.2 %–59.3 % | 0.94% | 1.20% |
| Mauritania 1 | (9.30° W, 19.40° N) | 48.3 %–65.6 % | 3.48% | 2.25% |
| Libya 1 | (13.35° E, 24.42° N) | 50.2 %–66.2 % | 2.51% | 2.25% |

Among the four PICS, Libya 4 was shown to be the most ideal site for the calibration, which is bright (the near-infrared [NIR] reflectance is high, at 55.3 %–60.6 %), most homogeneous (with mean spatial variation = 0.29 %), and most stable (with very low temporal variation = 0.81 %). On the other hand, similar interannual decline trends are given by the four PICS (Figure 12a). The NIR reflectance of Libya 4, Algeria 3, Mauritania 1, and Libya 1 declined by 16.21%, 17.57%, 17.20%, and 16.38% from 2007 to 2021, respectively. Therefore, it is reliable to fit the degradation of GOME-2A using Libya 4 site only.

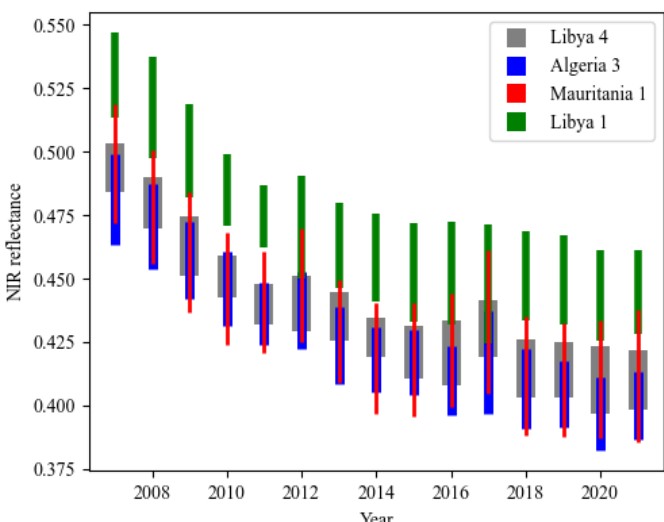

**Figure 12. Instrument degradation at four different calibration sites. Each bar shows the yearly average ± standard deviation.**

In addition, the degradation at different wavelengths may also differ. Degradation functions fitted by different wavelengths in the 735–758 nm are compared. A difference of less than 1% was found in the degradation from 2007 to 2021 fitted at different wavelengths (Figure 13a and Figure 13b). Figure 13b shows the variation of temporal decay at different wavelengths, indicating that inconsistency mainly occurs at the Fraunhofer line, which is inherently unstable in time. On the other hand, SIF retrieval relies on the filling of absorption lines. Extremely high fitting accuracy must be ensured if wavelengths are considered an influencing factor of the degradation function. Otherwise, the accuracy of

SIF retrieval will be greatly affected. Therefore, in this study, the wavelength dependence of the degradation within the 735–758 nm window is ignored.

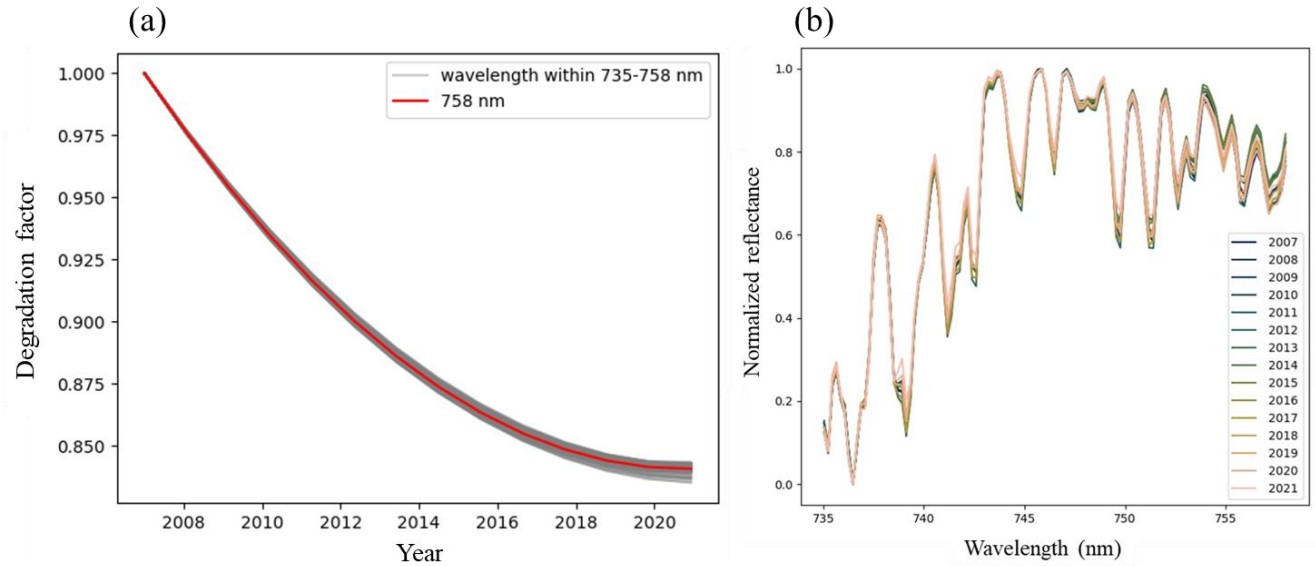

**Figure 13 (a) The degradation factor (Dfactor) fitted using reflectance at different wavelengths in a 735–758 nm fitting window. The red line is the result obtained at 758 nm, while the degradation functions fitted by other wavelengths were shown in gray. (b) Normalized NIR reflectance spectra in the 735–758 nm fitting window for different years in 2007–2021.**

### 5.2 Uncertainty in the temporal correction method

A wide range of radiance is essential for ensuring the representativeness of the temporal correction function, since the degradation may differ across different radiance levels. Although the pseudo-invariant sampling region selected in this study has a small spatial extent it has a large radiance range (48–284 mW m$^{-2}$ sr$^{-1}$ nm$^{-1}$), which almost covers that of the main vegetation areas at the near-infrared band (Figure 14); only the lowest value of vegetation radiance (24 mW m$^{-2}$ sr$^{-1}$ nm$^{-1}$) is not covered. Since the temporal invariance feature is required for the calibration site, it leaves a few optional

samples to choose from. The representativeness of samples may have an impact on the correction coefficient.

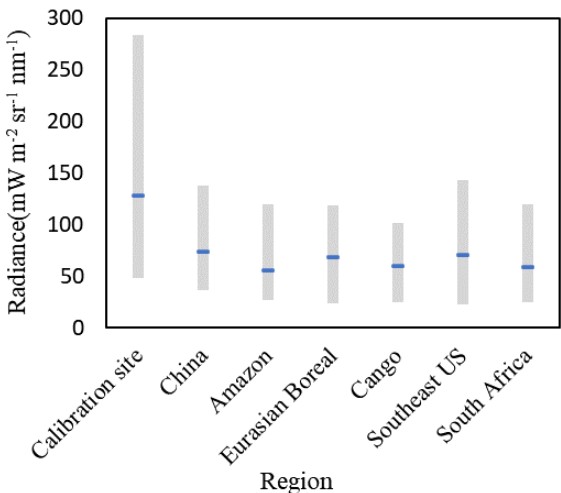

**Figure 14. Range of radiance at the near-infrared band at the calibration site and in the six main vegetated areas. The gray bars and blue lines are the range and mean of the datasets, respectively.**

The relative residuals of the corrected GOME-2A NIR radiance on vegetated targets under different radiance levels

were analyzed. As shown in Figure 15, the relative residuals are less than 20% when the NIR radiation is greater than 25

mW m$^{-2}$ sr$^{-1}$ nm$^{-1}$, and the averages of the relative residuals are less than 7%. The results indicate that the correction is basically accurate at different radiance levels. However, when the radiance is lower than 25 mW m$^{-2}$ sr$^{-1}$ nm$^{-1}$, the relative residual error reaches 40%. One reason for the result is that low radiance signals are greatly affected by random noise, resulting in poor comparability of GOME-2A and GOME-2C. Besides, the extremely low radiance level cannot be estimated

by the correction based on desert pixels. Therefore, the correction results can be inaccurate at pixels with low vegetation coverage or stressed vegetation.

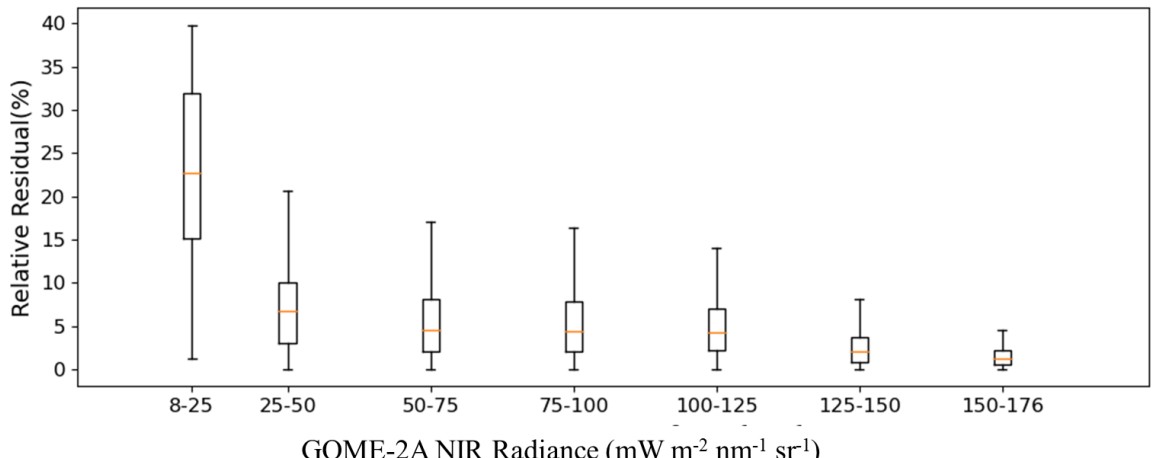

**Figure 15. Relative residual of NIR radiance (calculated as the absolute difference between GOME-2A and GOME-2C NIR radiance at the co-located points) at different radiance levels. Global vegetation targets with SIF signals greater than 0.1 mW m$^{-2}$**

**sr$^{-1}$ nm$^{-1}$ on July 1, 2019 were selected.**

Another limitation is that we only indirectly verify the reliability of the interannual trend of TCSIF when using several long-term remote sensing products, such as GPP, NDVI, and other SIF products. Direct validation data, such as field measurements, were not used to prove the accuracy of our results. In this respect, the huge discrepancy in scale between the

satellite SIF products (0.5°) and ground observations (<100 km$^2$) is one of the major obstacles. In fact, moreover, to our best knowledge, there is no decade-long in-situ SIF validation dataset available that is sufficiently reliable for such a direct validation, and the methodology of directly verifying satellite SIF based on in-situ measurements is still imperfect (Parazoo et al., 2019). The accuracy of TCSIF products needs to be verified via future applications.

Besides, the contamination of the lens may not be the only reason for GOME-2A's degradation. As shown in Figure 3,

the intra-annual variation in NIR reflectance does not decrease as the inter-annual average does. Instead, the intra-annual variation is growing with time. A similar phenomenon was found in the chlorine dioxide products (Pinardi et al., 2022) that GOME-2A results are noisier than those of GOME-2B, especially after 2011. These results suggest that in addition to the decline in reflectance over time caused by lens contamination, the temporal degradation is impacting GOME-2A measurements in other forms. However, the pattern of this effect is not clear now, further research is needed on the impact of

GOME-2A's degradation on its measurements in more aspects. Therefore, only the interannual decline trend was considered in this study, while the inevitable intra-annual variations caused by other factors such as the bidirectional reflectance distribution function and atmospheric scattering were neglected.

**5.3 Comparison with other long-term SIF products**

The annual average values of TCSIF and other long-term SIF products were compared (Figure 16). Importantly, most

of the long-term SIF products were in agreement, featuring an increasing trend of SIF from 2007 to 2018, except for NASA

SIF and SIFTER (Figure 16a–e). Among the temporal corrected SIF products, the annual curves of TCSIF and LT_SIFc* are generally consistent, while LT_SIFc* gives a higher growing trend of 1.247% yr$^{-1}$, and the uncertainty of the growing trend of TCSIF (0.15 % yr$^{-1}$) is lower. A slightly decreasing trend of -0.08 % yr$^{-1}$ characterized the SIFTER v2 product (Figure 16b), while the annual fluctuation of SIFTER v2 was clearly greatest among all the SIF products shown in Figure 16 (0.37 % yr$^{-1}$). The yearly trends according to TCSIF (1.06 % yr$^{-1}$) are close to the results from OCO-2 SIF from 2015 to 2021 (1.23% yr$^{-1}$, Figure 16f), while GOSIF shows a lower growing trend of 0.50% yr$^{-1}$ during the same period. Compared to GOSIF, which was derived from OCO-2 SIF using a machine learning method, TCSIF is even more consistent with OCO-2 SIF, suggesting the flaws of machine learning methods in maintaining the temporal trend of the original SIF products.

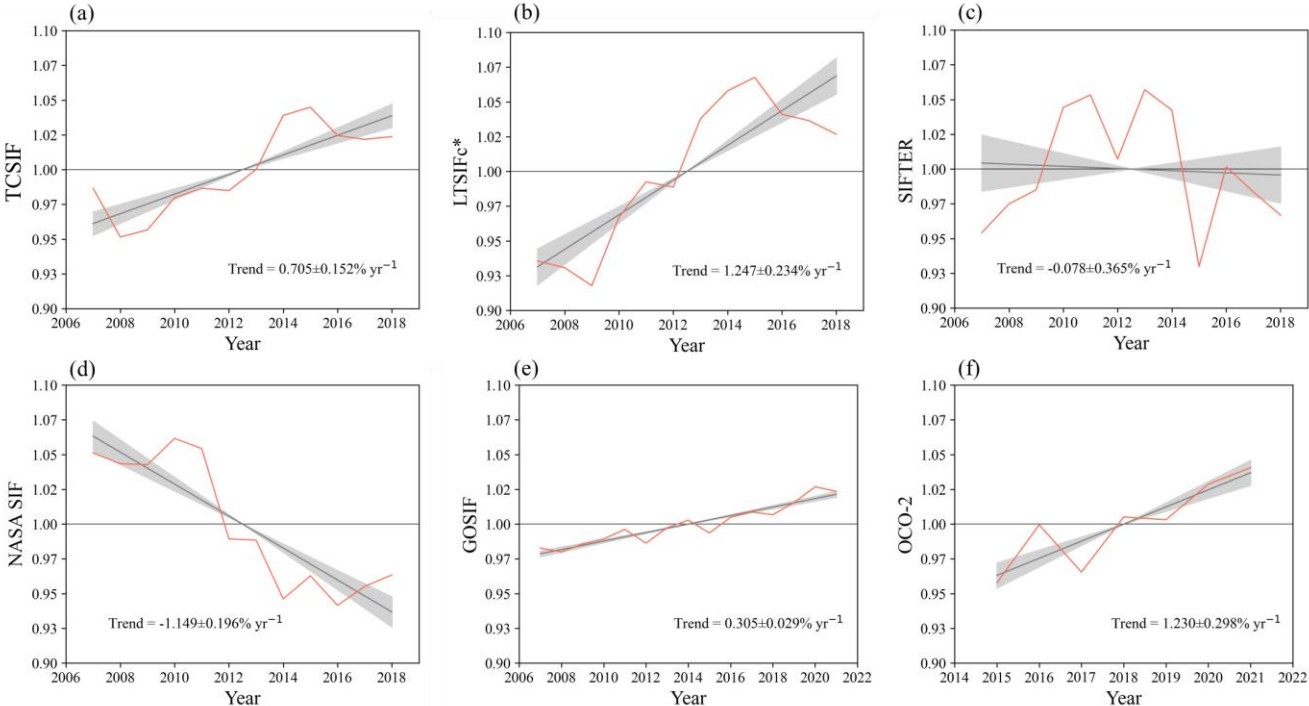

**Figure 16. Comparison of temporal trends in the annual SIF average from (a) TCSIF, (b) LT_SIFc*, (c) SIFTER v2, (d) NASA SIF during 2007–2018, as well as (e) GOSIF during 2007–2021, and (f) OCO-2 SIF during 2015–2021. All data shown are normalized to relative values (by dividing the mean). The shaded areas indicated the standard deviations.**

The large interannual fluctuation of SIFTER may be caused by the fact that its correction factor is seasonally based. No continuous correction functions were applied by SIFTER, which runs counter to the sensor's general pattern of temporal decay. (Lyapustin et al., 2014; Wang et al., 2012). In stark contrast, the least amount of interannual fluctuation was found in the GOSIF product. A neural network model was used for the spatiotemporal degradation of the GOSIF product, enabling GOSIF to inherit the time-stable signal from MODIS reflectance. However, this neural network model has been criticized for relying too much on training data such as reflectance data, and overlooking valuable information in the original observations (Ma et al., 2020). In the years not covered by the original OCO-2 SIF, the spatial distribution of GOSIF depends almost entirely on other input parameters of the data-driven model; hence, it cannot reliably capture the long-term temporal trend of SIF.

The LT_SIFc* product uses weak SIF signals over the Sahara Desert to fit the temporal decay pattern of the sensor, which can quickly generate corrected SIF products based on the monthly global maps provided by NASA SIF. Nevertheless, the method is not rigorous enough, since the sensor's degradation does not alter the SIF retrievals in a linear way. The post-processing steps, such as the zero-offset correction and quality-filtering procedures, will influence the distribution of global-gridded SIF products, leading to uncertainties arising in the correction function. Besides that, a large proportion of noise signals accompany the weak SIF signals over desert targets, thus restricting the fitting accuracy of the corrective function.

Meanwhile, LT_SIFc* is obtained by fusing three SIF products using the cumulative distribution frequency (CDF)-matching
approach. Accordingly, the spatiotemporal distribution of the original SIF signal may be forced to change due to adjustments
in the distribution frequency of each separate product. In this study, we corrected the degradation in radiance spectra rather
than SIF by using pseudo-invariant pixels over the Sahara Desert, which should provide a more reliable method.

To take advantage of SIF's ability to quickly capture changes in GPP, the temporal resolution of long-term SIF products
is supposed to be higher than 1 month and even a few days (Zhang et al., 2014; Zhang et al., 2016; Porcar-Castell et al.,
2014). However, LT_SIFc* cannot meet those temporal resolution requirements constrained by the original SIF products.
By contrast, the shorter, repeating cycle of GOME-2 was fully utilized in this study. Our work provides global daily level-2
SIF products that encompass the world's terrestrial area, which will greatly improve the application ability of global SIF
products for monitoring global vegetation dynamics.

## 5.4 Interannual trends for the TCSIF, NDVI, and GPP products

We compared the interannual trend of TCSIF with that of GPP and NDVI. Parameter values during the peak of the
growing season were compared to show the most lush period of vegetation each year. After spatial averaging of monthly
products, the yearly annual maximum values were calculated year by year.

As evinced by Figure 17a–e, the global yearly maximum of TCSIF showed a trend of increasing SIF intensity, which
was consistent with that of GPP and NDVI. The interannual fluctuation of TCSIF (0.16 %) slightly exceeded that of the
460 GPP and NDVI products (<0.1 % yr$^{-1}$) during the 2007–2021 period, and likewise for 2007–2016. The interannual trend
and associated uncertainty of each product are displayed in Figure 17f. Given that the timespan of Pmodel GPP stops at
2016, we selected the NDVI and TCSIF series from 2007 to 2016 for a fair comparison with Pmodel GPP, this is shown in
the bottom half of Figure 17f. Evidently, there are deviations in the interannual growing trend of vegetation when inferred
from different GPP products. For example, from 2007 to 2021, the interannual growth trend estimated by MODIS GPP
(0.64 %) surpassed that of TRENDY GPP (0.44 %). Meanwhile, the interannual growth rate of TCSIF was close to that of
MODIS GPP and Pmodel GPP in 2007–2021 and 2007–2016, respectively. Notably, when compared with the reflectance-
based index NDVI, the trend of TCSIF was more similar to that of GPP in both periods examined, indicating that TCSIF
was more capable of tracking GPP than NDVI.

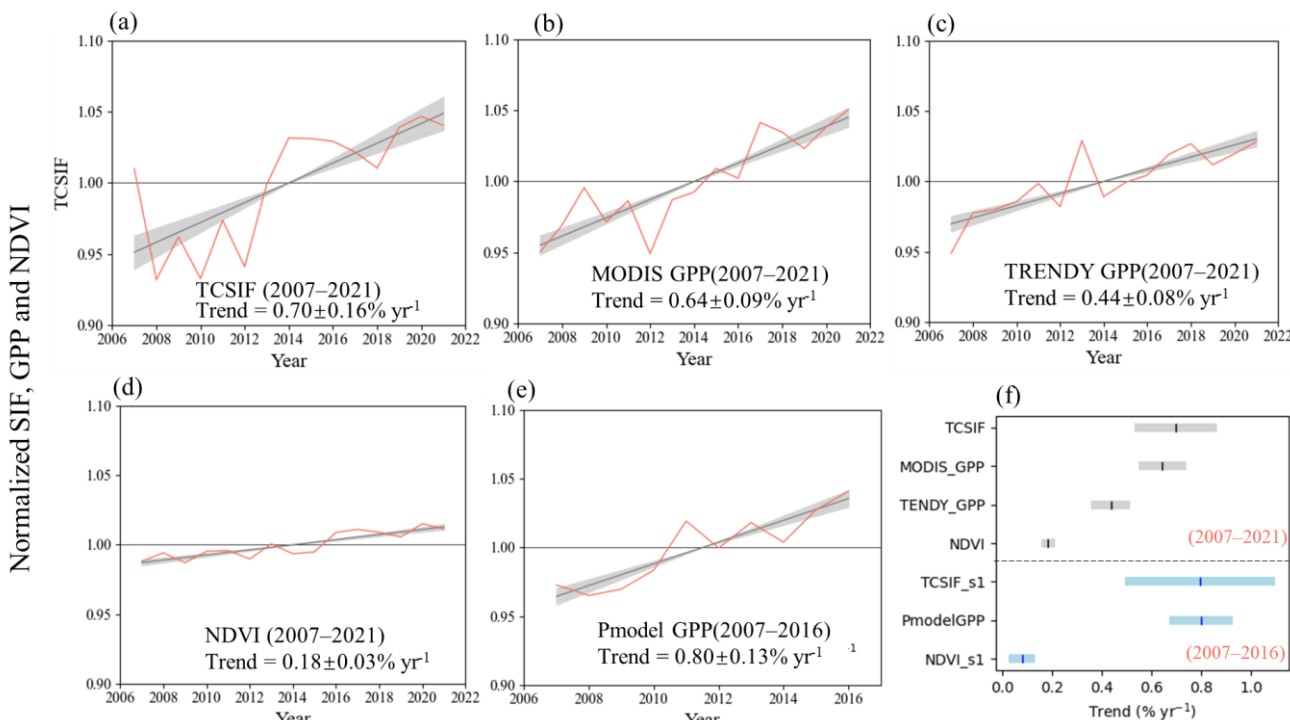

**Figure 17. Comparison of temporal trends in the yearly maximum from (a) TCSIF, (b) TRENDY GPP, (c) Pmodel GPP, (d) MODIS GPP, and (e) NDVI. All data shown are normalized to relative values (by dividing the mean). The shaded areas indicate the standard deviations and the gray lines represent the fitted lines which show the general trends. The interannual trends (shown by the gray or blue vertical short lines) of all the products and their uncertainties (shown by the blue or gray horizontal bars) are shown in (f). TCSIF_s1 and NDVI_s1 correspond to the TCSIF and NDVI series for 2007–2016.**

## 6 Conclusion

Degradation of the GOME-2A instrument has been a major barrier to producing consistent SIF products over an extended time period. By normalizing the instrument's degradation from 2007 to 2021, the radiance spectra of GOME-2A were successfully corrected. The calibrated GOME-2A NIR radiance was shown to be accurate by the comparison of GOME-2C, the mean bias is 1.85 mW $m^{-2}$ $sr^{-1}$ $nm^{-1}$. Based on the calibrated radiance, we were able to develop a temporally consistent SIF (TCSIF) dataset spanning decades for use in research. The TCSIF is strongly correlated with the NASA SIF, OCO-2 SIF, and TROPOMI SIF products in terms of its spatial distribution ($R^2 > 0.65$) and has a low retrieval residual (the RMS of residual is under 0.30 %). Our findings reveal that the TCSIF product yields a more reliable trend in vegetation SIF than does the GOME-2A dataset without a degradation correction applied. After undergoing the temporal correction, the vegetation SIF increased by 0.70 % per year during the 2007–2021 period, and 62.91 % of global vegetated regions saw an increase in their SIF, suggesting an overall increase in vegetation SIF and photosynthesis during the growing season. Compared with NDVI, the results obtained by TCSIF are closer to the GPP, indicating that the TCSIF product is a reliable proxy of vegetation activity.

We conclude that the TCSIF product developed in this study represents a significant advancement in our ability to accurately assess long-term changes in the SIF of vegetation on a global scale. This product can thus serve as a valuable reference for past and future studies of long-term SIF products and may provide important insights into the impact of climate change on vegetation photosynthesis.

## 7 Data availability

The global monthly GOME-2A SIF dataset (2007–2021) with correction of temporal degradation is openly available at https://doi.org/10.5281/zenodo.8242928 (See Table S1 for access to other related datasets). The corrected global GOME-2 SIF dataset can be obtained in two forms. The daily level 2 dataset is provided in hdf5 format. The name of these files is given as SIF_daily_YYYYMMDD.h5, in which YYYY, MM, and DD denote the year, month, and date, respectively. The level 3 datasets, which were aggregated monthly from the level 2 dataset, have a spatial resolution of 0.5° and are saved in TIFF format in chronological order, from 2007 to 2021. The name of these files is given as SIFpar_evi_monthly _YYYYMM.tif, in which SIF is the product type, par, and evi represent upscaled parameters, monthly denotes the temporal scale, and YYYY and MM are the year and month, respectively. The SIF output is stored in the hdf5 files along with other variables of interest for further processing and visualization. See the Appendix B for the structure of the hdf5 file.

## Appendix A. Supplementary material

**Table S1. Access to the dataset used to generate and compare TCSIF products.**

| Dataset Name | Description | Access |
| --- | --- | --- |
| GOME-2A/C Radiance | Level-1B product of GOME-2A and GOME-2C. | https://data.eumetsat.int/data/map/EO:EUM:DAT:METOP:GOMEL1 |
| Merra-2 PAR | Merra-2 meteorological assimilation reanalysis data (photosynthetically active radiation). | https://goldsmr4.gesdisc.eosdis.nasa.gov/data/MERRA2/M2T1NXRAD.5.12.4/ |

| MODIS MOD13C1 | MODIS Vegetation Indices 16-Day (Version 6.1). | https://lpdaac.usgs.gov/products/mod13c1v061/ |
|---|---|---|
| MODIS MOD43C4 | The MODIS Version 6.1 Nadir Bidirectional reflectance distribution Adjusted Reflectance (NBAR) product. | https://lpdaac.usgs.gov/products/mcd43c4v006/ |
| LT_SIFc* | Temporally corrected, global 0.05° level-3 SIF product. | https://doi.org/10.6084/m9.figshare.21546066.v1 |
| SIFTER | Level-2 daily GOME-2A SIF product accounts for biases. | https://www.temis.nl/surface/sif.php |
| NASA SIF | Level-2 daily SIF (at 740 nm) dataset from GOME-2A. | https://daac.ornl.gov/SIF-ESDR/guides/MetOpA_GOME2_SIF |
| GOSIF | A global, 0.05-degree product of solar-induced chlorophyll fluorescence derived from OCO-2, MODIS, and reanalysis data. | https://globalecology.unh.edu/data/GOSIF.html |
| OCO-2 SIF | Level-2 daily SIF (at 740 nm) dataset from OCO-2. | https://disc.gsfc.nasa.gov/datasets/OCO2_L2_Lite_SIF_10r/summary |
| TROPOMI SIF | Level-2 daily SIF (at 740 nm) dataset from TROPOMI. | ftp://fluo.gps.caltech.edu/data/tropomi/ |
| Trendy GPP | Global monthly 0.5° GPP based on the Dynamic Global Vegetation Model. | https://blogs.exeter.ac.uk/trendy/ |
| Pmodel GPP | Global daily 0.5° GPP based on a LUE model (P-model). | https://zenodo.org/records/1423484 |
| MODIS GPP | 8-day composite, 500 m GPP product product based on the radiation use efficiency concept. | https://lpdaac.usgs.gov/products/mod17a2hv061/ |

**Appendix B. Level 2 file description**

The fields of the level 2 products include:

(1) SIF retrievals: including the instance SIF retrieved using the data-driven algorithm (SIF_740), the day-length corrected SIF (SIF_daily), and the relative error estimations (the 1-$\sigma$ error (sigma_1), $\chi^2$ and the quality assurance field(QA)).

(2) Geo Locations, fields that describe the location: including the latitude and longitude of the center and the boundary of each footprint, the solar and viewing angles.

(3) Ancillary data: including the reflectance at red (ps_red) and far-red bands(ps_NIR), cloud fraction, the mean radiance in the 735–758 nm fitting window (Rad_NIR), and NDVI calculated from GOME-2 reflectance.

**Author contribution.** LL and XL designed the experiments and CZ carried them out. CZ and SD developed the model code and generated the products. CZ prepared the manuscript with contributions from all co-authors.

**Competing interests.** The authors declare no conflict of interest.

**Disclaimer.**    Publisher's note: Copernicus Publications remains neutral with regard to jurisdictional claims in published maps and institutional affiliations.

**Acknowledgments.** This work was supported by the National Key Research and Development Program of China, grant number 2022YFB3904801, and the National Natural Science Foundation of China under Grant 42071310. The authors
acknowledge P. Kohler and C. Frankenberg for publicly providing TROPOMI SIF and OCO-2 SIF datasets. We acknowledge S. Sitch for sharing the link to the latest version of TRENDY GPP. CZ acknowledges SH Wang for the instruction on the algorithm of LT_SIFc* data.

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
