# Peer review of "TCSIF: A temporally consistent global GOME-2A solar-induced fluorescence dataset chlorophyll with correction of sensor degradation"

_Earth System Science Data, 2023_

## Author Comment (AC1)

https://doi.org/10.5194/essd-2023-329

**Response to Reviewer 1 Comments:**

Zou et al. corrected the degradation trend of GOME-2A SIF with a pseudo-invariant method, based on a 1 degree x 1 degree calibration region in the Sahara Desert. Unlike previous studies, the correction in this study was conducted at radiance level and daily basis, using a fitted quadratic polynomial function. Then the SIF signal was retrieved using a data-driven algorithm, which was further scaled to monthly average with a PAR-based upscaling model. The global trend of the corrected SIF data was also compared with several existing SIF/GPP/NDVI products. The study is overall nicely conducted.

Thank you for your thorough and insightful review of our manuscript. We appreciate the time and effort you invested in providing valuable feedback. We have carefully considered each of your comments and suggestions. Please find the point-by-point responses below.

**Major comment 1: Benchmarks used for trend evaluation:** The authors employed MODIS NDVI and a number of GPP datasets from MODIS and model simulations, to evaluate the trend of TCSIF. However, MODIS NDVI has a known saturation effect, which would especially influence the evaluation of annual maximum (Fig. 11). MODIS GPP and TRENDY GPP are also known to have issues with their trends (Anav et al., 2015, https://doi.org/10.1002/2015RG000483). I would not trust the evaluation against these datasets.

**Response:** Thanks for this comment. We agree that the trend of GPP and NDVI cannot serve as benchmarks for the time series of SIF, in this study, they can only used for indirect comparison. We used the annual maximum to derive trends when the vegetation is most vigorous, while NDVI shows its limitation of not being able to catch the trend of GPP compared with SIF.

These indirect validations were moved to the discussion part (Sec.5.3 and Sec. 5.4). Instead, we added a two-step validation for both the corrected radiance spectra (in Sec. 4.1) and the retrieved SIF (in Sec. 4.3) of the revised manuscript.

Radiance spectra obtained from GOME-2C serve as a benchmark for the radiance spectra. Being a sensor that measures the same bands with the same spectral resolution as GOME-2A, GOME-2C has a later launching time in November 2018. Thus, measurements at the initial launch stage of GOME-2C can be taken as accurate values that are not affected by degradation. The pseudo-invariant calibration method was verified by validating temporal corrected radiance in Sec. 4.1 **(Line 256)**:

**4.1 Correction of GOME-2A sensor degradation**

The temporally corrected GOME-2A NIR radiance was validated using GOME-2C radiance spectra (Figure 4). For the corrected GOME-2A radiance, the scatter plot shows that the majority of points are

https://doi.org/10.5194/essd-2023-329

concentrated near the 1:1 line (Figure 4a). The slight positive offset of the linear regression may be caused by the lower orbit height of GOME-2C (817 km) than GOME-2A (827 km), which results in less atmospheric absorption in GOME-2C. The difference between the two products followed a Gaussian distribution with a small mean value of 1.85 mW m$^{-2}$ sr$^{-1}$ nm$^{-1}$, which is 2.3% of the mean GOME-2A radiance (Figure 4b). On the contrary, the mean deviation without temporal correction is 15.16W m$^{-2}$ sr$^{-1}$ nm$^{-1}$ (Figure 4d).

[Figure]

**Figure 4 Comparison between GOME-2A and GOME-2C NIR radiance after (a,b) and before (c,d) the temporal correction on 1 July 2019. The histogram of difference for GOME-C NIR radiance minus the corrected and uncorrected GOME-2A NIR radiance was shown in (b) and (d), respectively. Spatially matched pixels with cloud fraction of lower than 0.3 and SZA of lower than 70° were selected.**

Besides, TCSIF has been validated in Sec. 4.3 **(Line 295)** with OCO-2 SIF and TROPOMI SIF, the results are shown below:

**4.3 Spatial distribution of the TCSIF dataset**

OCO-2 SIF and TROPOMI SIF were also involved in the validation of TCSIF (Figure 7 e,f). To avoid discrepancies in wavelength and the overpassing time, the day-length corrected 740 nm provided by OCO-2

https://doi.org/10.5194/essd-2023-329

SIF, TROPOMI SIF, and TCSIF were compared. The spatially matched points were selected. TCSIF versus OCO-2 SIF and TCSIF versus TROPOMI SIF comparisons were conducted in July 2019 and July 2021, respectively. Both comparisons show high consistency with $R^2 > 0.65$, and the linear regression results are close to the 1:1 line.

[Figure]

Figure 7. Comparison of TCSIF vs. NASA SIF on 14 January (a) and 15 July (b) in the year 2008, 15 July 2017 (c), and 14 January 2018 (d). **Comparison of TCSIF versus OCO-2 in July 2019 (e) and TCSIF versus TROPOMI SIF in July 2021 (f).** The comparison was made based on the level 2 product. Co-located pixels over land with a cloud fraction < 0.3 have been selected. The color of the scatter points represents the density of the points. The blue dotted line and the black solid line represent the line fitted based on the scatter points and the 1:1 line, respectively.

Besides, temporal comparisons were made between TCSIF and OCO-2 by demonstrating the trend map (detailed in **minor comment 8**) as well as the yearly average curve (detailed in **minor comment 9**). The

https://doi.org/10.5194/essd-2023-329

analysis of the TROPOMI time series is not shown because the TROPOMI dataset only contains gridded data between 2018.4 and 2021.3, which means that there are only two full years of data in 2019 and 2020. Therefore, TROPOMI SIF is currently not able to provide accurate temporal trends.

**Major comment 2: Pseudo-invariant method and site:** The authors employed a pseudo-invariant method and selected a 1 degree x 1 degree non-vegetated region which trend was taken as the temporal degradation in the GOME-2A instrument. It has a strong underlying assumption that the degradation trend is the same for different locations and different radiance levels. It remained to be tested/discussed to what extent this is true.

**Response:** Thanks for this comment. The influence of BRDF and atmospheric conditions limits the selection of diversified pseudo-invariant sample points. Only bright, stable, and homogenous sites can be chosen for the calibration of GOME-2A, thus the desert surface is the most ideal choice. Since the decay function fitted by only one site may not be convincing enough, we added three other commonly used pseudo-invariant calibration sites (PICS). The results show high consistency between degradation functions fitted by those PICS. We have added relative analysis to the discussion part of the revised manuscript as follows:

Line 346:

**5.1 Degradation at different locations and wavelengths**

In this study, only one calibration site (Libya 4) was used for the fitting of the degradation function. The results may be different for different sites. Previous studies have compiled 20 pseudo-invariant calibration sites(PICS) for instrument calibration (Cosnefroy et al., 1996; Bacour et al., 2019). We have involved three other commonly used sites, and the relative information is shown in Table 3.

**Table 3 Four pseudo-invariant calibration sites (PICS) and related information**

| Site name | location | NIR reflectance | mean variation | spatial | temporal variation |
|---|---|---|---|---|---|
| Libya 4 | (23.00° E, 29.00° N) | 55.3 %–60.6 % | 0.29 % | | 0.81 % |
| Algeria 3 | (7.66° E, 30.33° N) | 49.2 %–59.3 % | 0.94% | | 1.20% |
| Mauritania 1 | (9.30° W, 19.40° N) | 48.3 %–65.6 % | 3.48% | | 2.25% |
| Libya 1 | (13.35° E, 24.42° N) | 50.2 %–66.2 % | 2.51% | | 2.25% |

Among the four PICS, Libya 4 was shown to be the most ideal site for the calibration, which is bright (the near-infrared [NIR] reflectance is high, at 55.3 %–60.6 %), most homogeneous (with mean spatial variation = 0.29 %), and most stable (with very low temporal variation = 0.81 %). On the other hand, similar

https://doi.org/10.5194/essd-2023-329

interannual decline trends are given by the four PICS (Figure 12 a). The NIR reflectance of Libya 4, Algeria 3, Mauritania 1, and Libya 1 declined by 16.21%, 17.57%, 17.20%, and 16.38% from 2007 to 2021, respectively. Therefore, it is reliable to fit the degradation of GOME-2A using Libya 4 site only.

[Figure]

**Figure 12. Instrument degradation at four different calibration sites. The center of each bar shows the yearly average, while the length of the bars represents twice the yearly standard deviation.**

Results from the four PICS show that there is no significantly greater or smaller degradation at the Libya 1 site although the radiance level is higher than the other three sites.

Various targets with very high or very low radiance levels, such as snow or water bodies, were not selected in this study. This is limited by the temporal stability requirement of choosing the pseudo-invariant calibration site. Regarding this, we stated in the discussion that the radiance range of the Libya 4 site almost covers the radiance levels of vegetation. Thus, the degradation function fitted by these sites be considered effective for vegetation targets. The relative statement is as below. Besides, the sentence *"A wide range of radiance is essential for ensuring the representativeness of the temporal correction function, since the signal-to-noise ratio may differ across different radiance levels."* in line 376 of the previous manuscript was not accurate, and the "signal-to-noise ratio" was changed to "degradation" in the revised manuscript.

==**Line 375:**==

5.2 Uncertainty in the temporal correction method

A wide range of radiance is essential for ensuring the representativeness of the temporal correction function, since the **degradation** may differ across different radiance levels. Although the pseudo-invariant sampling region selected in this study has a small spatial extent it has a large radiance range (48–284 mW $m^{-2}$ $sr^{-1}$ $nm^{-1}$), which almost covers that of the main vegetation areas at the near-infrared band (Figure 14); only the lowest value of vegetation radiance (24 mW $m^{-2}$ $sr^{-1}$ $nm^{-1}$) is not covered. Since the temporal

https://doi.org/10.5194/essd-2023-329

invariance feature is required for the calibration site, it leaves a few optional samples to choose from. The representativeness of samples may have an impact on the correction coefficient.

Another limitation is that we only indirectly verify the reliability of the interannual trend of TCSIF when using several long-term remote sensing products, such as GPP, NDVI, and other SIF products. Direct validation data, such as field measurements, were not used to prove the accuracy of our results. In this respect, the huge discrepancy in scale between the satellite SIF products (0.5°) and ground observations (<100 km$^2$) is one of the major obstacles. In fact, moreover, to our best knowledge, there is no decade-long in-situ SIF validation dataset available that is sufficiently reliable for such a direct validation, and the methodology of directly verifying satellite SIF based on in-situ measurements is still imperfect (Parazoo et al., 2019). The accuracy of TCSIF products needs to be verified via future applications.

[Figure]

**Figure 14. Range of radiance at the near-infrared band at the calibration site and in the six main vegetated areas. The gray bars and blue lines are the range and mean of the datasets, respectively.**

For example, in Fig. 9, the authors showed the difference in the temporal trend with and without the correction, which is basically the degradation trend. There are large spatial variabilities in the degradation trend.

**Response:** Thanks for the comment. It's true that we can see large spatial variabilities in the degradation trend between NASA SIF and TCSIF in Fig.9. However, there will be uncertainties if we take the difference between NASA SIF (without correction) and TCSIF (with correction) in Fig. 9 as the degradation trend of GOME-2. Firstly, different fitting windows and data-driven models were used for SIF retrieval by NASA SIF and TCSIF, which caused inevitable differences in the spatial distribution between the two products. Secondly, the zero-offset corrections were conducted for both of the products to get the final global SIF map. The zero-offset correction deals with the systematic error of the SIF retrieval variant with latitude and radiance. The correction coefficient changes with latitude, radiance, and time, so there may be great uncertainty if taking the difference in the growth trend of the two products as a recession trend.

https://doi.org/10.5194/essd-2023-329

**Minor comment 1:** Line 13: This is not accurate. As the authors cited in Table 2, there have been several efforts for correcting the temporal degradation of GOME-2A SIF.

**Response:** Thanks for this comment. Here, we want to emphasize that the existing GOME-2 temporal correction SIF products still have unfixed problems. For example, LTSIFc is directly corrected based on the 0.5° gridded global SIF products, but the degradation of the sensor is not linearly transmitted to the degradation of SIF products. Meanwhile, SIFTER gives quarterly correction factors, leading to large annual fluctuations. Therefore, it is very necessary to solve the temporal degradation problem from the radiance source. The sentence has been revised as follows:

However, serious temporal degradation of the GOME-2A instrument is a problem, **and for now, there is a lack of time-consistent GOME-2A SIF products that meet the needs of temporal trend analysis.**

**Minor comment 2:** Line 19: By "weather conditions", I assume it referred to only light conditions?

**Response:** Thanks for this comment, in this manuscript, by "weather conditions" we mean that the variance of atmospheric conditions (for example, the atmospheric scatter and the effect of clouds) during the day was considered, not only light conditions caused by solar angles, by using PAR instead of cos(SZA). Using this approach, it's more accurate to upscale instantaneous clear-day SIF to the daily average.

**Minor comment 3:** Line 88: MCD43C4 is provided at daily resolution.

**Response:** We apologize for the mistake, MCD43C4 is daily data synthesized over 16 days, sorry for confusing the two concepts in the previous version of the manuscript. The description was revised as follows: The MODIS Version 6.1 Nadir Bidirectional reflectance distribution Adjusted Reflectance (NBAR) product (MCD43C4) (Schaaf et al., 2002) records the surface reflectance at a nadir viewing angle for each pixel at local solar noon. It has a spatial resolution of 0.05° × 0.05° and a **daily** temporal resolution (Schaaf et al., 2002). The MODIS NBAR product is considered stable over long periods of time and was used here to investigate the homogeneity and stability of the calibration site (see Sect. 3.1).

**Minor comment 4:** Line 112: The authors did not provide the links for the datasets used in this study. For NASA SIF, there is a recent updated version here: https://daac.ornl.gov/SIF-ESDR/guides/MetOpA_GOME2_SIF.html. It is generated with updated GOME-2 Level 1B radiances and irradiances, and I heard that the degradation trend has been largely corrected. If the authors used an older version, I'd suggest comparing with this updated dataset too.

Thanks for this comment. We are already using the latest version of the NASA SIF data. Although this

https://doi.org/10.5194/essd-2023-329

version of the data uses the latest GOME-2 Level 1b data, as explained in the user guide of the latest NASA SIF product, the data has not yet been analyzed for potential errors caused by instrument degradation trends and therefore, is not recommended for long-term analysis.

We have added the link and the introduction of NASA SIF, as well as other products used in the dataset fraction of the revised manuscript as follows:

**Line 113:**

Next, we selected four long-term SIF products spanning more than one decade for comparison, including the LT_SIFc*(1995–2018) (Wang et al., 2022), SIFTER v2 (2007–2018) (Schaik et al., 2020), GOSIF (2000–2022) (Li and Xiao, 2019), and GOME-2 SIF products generated by the National Aeronautics and Space Administration (hereon abbreviated as NASA SIF, https://daac.ornl.gov/SIF-ESDR/guides/MetOpA_GOME2_SIF) (2007–2018) (Joiner et al., 2013&2016). The LT_SIFc*(**https://doi.org/10.6084/m9.figshare.21546066.v1**) is a data fusion product of GOME, SCIAMACHY, and GOME-2, having a spatial resolution of 0.05° and a temporal resolution of 1 month. It dealt with the temporal decay of the instrument based on statistics of SIF signals in the Sahara Desert. The SIFTER v2 product **(https://www.temis.nl/surface/sif.php)** is the point-by-point SIF product retrieved from GOME-2 measurements after applying a time-related correction factor; it was composited to yield a 0.5°, monthly global map in this study. The GOSIF product **(https://globalecology.unh.edu/data/GOSIF.html)** is the spatiotemporal expansion product based on the global neural network model and OCO-2 SIF V8r product, with a spatial resolution of 0.05° and a temporal resolution of 8 days. Apart from the SIF products spanning decades, the OCO-2 SIF product **(https://disc.gsfc.nasa.gov/datasets/OCO2_L2_Lite_SIF_10r/summary)** from 2015 to 2021, **and TROPOMI SIF (ftp://fluo.gps.caltech.edu/data/tropomi/) from 2018 to 2021** are also included here for comparative purposed given its high accuracy and it is being less affected by sensor degradation. All SIF products were resampled to a 0.5°, monthly spatiotemporal resolution and were compared with TCSIF to assess long-term trends in this study. Additionally, we used the NASA GOME-2A level 2 SIF product, which has not been corrected for temporal decay, to verify the spatial distribution of our product. Key information about these SIF products is presented in Table 2.

**Minor comment 5:** Line 159: The function f should be introduced here.

**Response:** Thanks for this comment. In section 3, we mainly introduce the method of fitting the time degradation. The specific fitting coefficients are given in section 4. According to your suggestions, we show the form of the temporal-degradation function in section 3 **(Line 166)** and guide the readers in finding the fitting process in the results section **(Line 253).** The modifications are as follows:

https://doi.org/10.5194/essd-2023-329

**Line 166:**

Thus, a time-dependent correction factor was calculated, and the temporal correction function was assumed to be a second-order polynomial as follows:

$$\text{Dfactor} = a \cdot NOD^2 + b \cdot NOD + c, \tag{1}$$

where Dfactor is the normalized correction factor describing the temporal degradation. NOD is the number of elapsed days since January 1st, 1900, starting with 1. a, b, and c are the fitting coefficients of the polynomial function based on the near-infrared radiance of the pseudo-invariant site, the detailed analysis can be found in Sect.4.1.

**Line 253:**

**The second-order polynomial (Dfactor) fitted by the normalized coefficient was used in Eq. (2) to calibrate the instrument's degradation since 1 January 2007**, as given by:

$$\text{Dfactor(NOD)} = 80.298 \times \left(\frac{NOD}{100000}\right)^2 - 70.123 \times \frac{NOD}{100000} + 16.142, \ (R^2 = 0.851), \tag{11}$$

where Dfactor is the degradation coefficient and "NOD" is the number of days since January 1st, 1900.

**Minor comment 6:** Line 217: "upscale the daily observations to monthly values" – This is confusing. The previous sentence emphasized the potential bias of upscaling instantaneous measurements to daily values.

**Response:** Thanks for the comment. The statement here is to illustrate that compared with the upscaling method using cos(SZA), extending the instantaneous SIF to the daily average using PAR has a lower potential bias. We have modified this sentence to be more understandable in the revised manuscript:

**Line 224:**

In previous studies, the global satellite-observed SIF was upscaled to a daily scale by using the diurnal cycle of the cosine of the solar zenith angle (cos[SZA]) to correct for day-length effects (Frankenberg et al., 2011; Zhang et al., 2018). These effects can cause large overestimates of SIF on cloudy days because the satellite-observed SIF data are only available on clear-sky days. **In this study, the downwelling PAR rather than cos(SZA) was used to compensate for the significant effects of diurnal weather changes due to cloud and atmospheric scattering (Hu et al., 2018) while upscaling the instantaneous SIF to monthly values.** The all-sky monthly averaged SIF ($\text{SIF}_{\text{mon}}$) can be determined using the PAR-based upscaling model, as follows:

$$\text{SIF}_{\text{mon}} = \begin{cases} \frac{\sum_{\text{mon}}^{M} \text{SIF}_{\text{ins}}}{\sum_{\text{mon}}^{M} \text{PAR}_{\text{ins}} \times \text{EVI}_{\text{ins}}} \times \text{PAR}_{\text{mon}} \times \text{EVI}_{\text{mon}}, \text{if } \text{EVI}_{\text{mon}} > 0.2 \\ \\ \frac{\sum_{\text{mon}}^{M} \text{SIF}_{\text{ins}}}{\sum_{\text{mon}}^{M} \text{PAR}_{\text{ins}}} \times \text{PAR}_{\text{mon}}, \text{if } \text{EVI}_{\text{mon}} \leq 0.2 \end{cases}, \tag{10}$$

https://doi.org/10.5194/essd-2023-329

**Minor comment 7:** Line 233: Could you clarify what the "normalized coefficients" stand for?

**Response:**

Thanks for the comment. The "normalized coefficients" are calculated by dividing the NIR reflectance by the value of its fitting function (shown in Figure 3a) on the starting date (1 January 2007). This ensures that the second-order correction function (Dfactor), which is fitted based on the normalization coefficient, has a value of 1 on the starting date. We have to apologize that the previous Figure 3 (b) was wrongly depicted, which might be confusing, we have revised it in the latest version of our manuscript. In fact, there is only a multiplicative factor between the reflectivity and the normalization coefficients, and the distribution of the scatter points is exactly the same. We have added explanations to both the figure captions and the main text, and the revisions are as follows:

**Line 240:**

[Figure]

Figure 3. Temporal variation in the TOA reflectance at 758 nm at the calibration site (left) and the temporal correction coefficients used to compensate for the degradation of GOME-2A since 2007 (right). **The blue dots and red curves in (a) represent NIR reflectance and the fitted quadratic function. The normalized coefficients in (b) were calculated by dividing the NIR reflectance by the value of the quadratic function in (a) at the starting date (1 January 2007).** "NOD" in the degradation correction equation is the GOME-2A acquisition date since 2007, which equals the number of days from 1 January 1900.

A second-order polynomial was fitted to describe the temporal degradation in the reflectance signal of GOME-2A. Figure 3 a illustrates the temporal variation in TOA reflectance at 758 nm at the calibration site. Significant and continuous degradation can be observed; however, this nonlinear trend could be accurately captured by a quadratic polynomial function with a determination coefficient ($R^2$) of 0.851. These results

https://doi.org/10.5194/essd-2023-329

indicated that, overall, the GOME-2A instrument degraded by 16.21 % from 2007 to 2021. This temporal degradation was considered spectrally constant in the narrow fitting window of SIF retrieval (735–758 nm).

**By dividing the NIR reflectance by the value of the fitted function at the starting date (1 January 2007), we obtain the normalized temporal correction coefficients**, as shown in Figure 3 b. The second-order polynomial (Dfactor) fitted by the normalized coefficient  was used in Eq. (2) to calibrate the instrument's degradation since 1 January 2007, as given by:

$$\text{Dfactor(NOD)} = 80.298 \times \left(\frac{\text{NOD}}{100000}\right)^2 - 70.123 \times \frac{\text{NOD}}{100000} + 16.142, \ (R^2 = 0.851), \tag{11}$$

where Dfactor is the degradation coefficient and "NOD" is the number of days since January 1st, 1900.

**Minor comment 8:** Fig. 10: The two green colors are not very distinguishable visually. Also, comparison with other datasets in terms of the trend maps may be useful, e.g., which regions have significant increase/decrease, are they consistent among different datasets?

**Response:** Thanks for this comment, the color of the "increase" trend is changed to be more distinguishable as below:

**Line 328:**

[Figure]

**Figure 10. Map of trends in the annual average GOME-2A SIF for 2007–2021. The inset shows the percentage of areas characterized by four types of trends (significant increase: positive correlation with $p < 0.05$; increase: positive correlation with $p \geq 0.05$; decrease: negative correlation with $p \geq 0.05$; significant decrease: negative correlation with $p < 0.05$).**

https://doi.org/10.5194/essd-2023-329

Besides, the trend maps of GPP, NDVI, NASA SIF, and OCO-2 SIF were analyzed in the revised manuscript as follows:

**Line 332:**

[Figure]

**Figure 11. Map of trends in the annual average (a)NASA SIF for 2007–2018, (b) OCO-2 SIF for 2015–2021, as well as (c) Trendy GPP and (d) NDVI for 2007–2021. The colors represent four types of trends (significant increase: positive correlation with $p < 0.05$; increase: positive correlation with $p \geq 0.05$; decrease: negative correlation with $p \geq 0.05$; significant decrease: negative correlation with $p < 0.05$).**

As shown in Figure 11, 57.11% of vegetation areas are facing a decline in NASA SIF. On the opposite, as seen from OCO-2 SIF, trendy GPP, and NDVI, vegetation is growing over a large area globally (>70%) from 2007 (or 2015) to 2021. The main inconsistency between NASA SIF and the other products occurs in central and southern Africa, eastern Europe, and southern North America, where NASA SIF declines and the others increase. In southeastern China, vegetation greenings were found by TCSIF, OCO-2 SIF, and NDVI, while an insignificant downward trend was shown by trendy GPP. Vegetation growth in southern North America, Europe, the Amazon rainforest, central Africa, and Southeast Asia was detected by all the products apart from NASA SIF.

**Minor comment 9:** Line 311: Please justify why the trend of annual maximum was selected for evaluation, not annual mean or annual minimum.

https://doi.org/10.5194/essd-2023-329

**Response:** Thanks for this comment. For comparison between different vegetation parameters (SIF, GPP, and NDVI), the annual maximum was selected because the annual maximum captures the period of peak vegetation activity, typically coinciding with the peak of the growing season. This allows us to focus on the most dynamic phase plant growth. Meanwhile, during the season with the highest SIF intensity, the effect from cloud and atmospheric scatter, as well as spatial noise caused by the retrieval algorithm are relatively low. On the contrary, during the period with the minimum SIF, the low global average signal is more susceptible to interference from noise and weather conditions, thus affecting the accuracy of the trend fitted. We have explained why and how we calculate the maximum value in the revised manuscript as shown below:

**Line 441:**

*5.4 Interannual trends for the TCSIF, NDVI, and GPP products*

[revised manuscript text omitted]

**Minor comment 10:** Fig. 12: it is also worth noting that the year-to-year variations are very different among different SIF products. Any idea why? Could you also add NASA SIF here to see if TCSIF has consistent year-to-year variations with NASA SIF?

**Response:** Thanks for this comment. It is true that we can see different year-to-year variations for different products, even though they are generated using the same sensor. Although all rely on GOME-2A measurements, the year-to-year variations shown by NASA SIF, TCSIF, and SIFTER are not always consistent. The discrepancy can be caused by the difference in a retrieval algorithm, post-processing method, and temporal correction method. For example, different retrieval model was used for TCSIF (SVD linear model) and NASA SIF (PCA non-linear model), and different temporal correction methods were applied for TCSIF (using daily correction factors) and SIFTER (using seasonal correction factors). According to minor comment 9, annual averages were selected for all the comparisons between SIF products. Besides, the interannual variation of NASA SIF was also added in the revised manuscript. The comparison of different SIF products as well as the analysis for the discrepancy were shown in the discussion (Sec. 5.3) in the revised manuscript. Detailed results are shown in **minor comment 9**.

Comparison using annual averages demonstrates better consistency in inter-annual fluctuations between TCSIF and other products. Especially, the interannual variation given by the annual mean of LTSIFc* shows a clearer trend ($1.25 \pm 0.23\%$ yr$^{-1}$) compared to that fitted by the annual maximum in the

https://doi.org/10.5194/essd-2023-329

previous version of the manuscript (0.21 $\pm$0.17% yr$^{-1}$). Meanwhile, the general conclusion was similar to that we derived in the previous version of the manuscript. For example, NASA SIF intensity declined from 2007 to 2018, and SIFTER shows a near-constant interannual trend with large yearly fluctuation, while overall increasing trends are captured by other SIF products.

**Editorial suggestions:**

Thanks for the advice, the expressions have been corrected as follows:

Line 41: "Given" -> "Given that"

**Given that** its global coverage capability starts from 2007, the GOME-2 satellite-based SIF dataset has been the most widely used for global monitoring of GPP, crop yield, drought, vegetation phenology, etc.

Line 52: "Forest" -> "forests"

For example, Yang et al. (2018) reported the SIF emission of the Amazon **forest** decreased during the 2015/2016 El Niño event when analyzed using the GOME-2 SIF data by Joiner et al. (2016), which is in conflict with the increase of the enhanced vegetation index (EVI) and downward solar shortwave radiation.

Line 116: "expansion" -> "extrapolation"

Line 122:

The GOSIF product (https://globalecology.unh.edu/data/GOSIF.html) is the spatiotemporal **extrapolation** product based on the global neural network model and OCO-2 SIF V8r product, with a spatial resolution of 0.05° and a temporal resolution of 8 days.

Bacour, C., Briottet, X., Bréon, F.-M., Viallefont-Robinet, F., and Bouvet, M.: Revisiting Pseudo Invariant Calibration Sites (PICS) Over Sand Deserts for Vicarious Calibration of Optical Imagers at 20 km and 100 km Scales, Remote Sensing, 11, 1166, 2019.

Cosnefroy, H., Leroy, M., and Briottet, X.: Selection and characterization of Saharan and Arabian desert sites for the calibration of optical satellite sensors, Remote Sensing of Environment, 58, 101-114, https://doi.org/10.1016/0034-4257(95)00211-1, 1996.

---

## Author Comment (AC2)

https://doi.org/10.5194/essd-2023-329

**Response to Reviewer 2 Comments:**

Zou et al. explored a pseudo-invariant method to resolve the temporal degradation issue with GOME-2A data and tested how the corrected SIF product improved. The study is of great significance in providing more reliable long-term SIF data. The manuscript was well written and the messages were well delivered. See my detailed comments below.

Thank you for taking the time to provide a thorough and insightful review of our manuscript. We appreciate your effort in providing valuable feedback. We carefully considered each of your comments and suggestions. Please find the point-by-point responses below.

**Major comment 1:** Not enough information has been given regarding why the GOME-2A is subject to degradation and where it occurred except in lines 43-44 that "GOME-2A is an optical spectrometer that measures reflected sunlight and is therefore sensitive to instrument degradation". If degradation is a problem for all the optical spectrometers, it should also be a problem for MODIS VIS/NIR bands. Identifying the reason for the degradation is crucial for determining the correction method to apply. The correction method applied in the present study assumes the GOME-2A radiance is uniformly downscaled at all wavelengths. This would not be correct if the degradation is caused by a dirty lens. Some results and discussions are required to validate this assumption.

**Response:** Thanks for this comment. We have investigated the reason for GOME-2's degradation in more detail. ESA's inspection of the GOME-2 sensor degradation indicates that the most likely cause is contamination of the optical lens (A. Hahne et al.,2012). The contamination was mainly caused by the Arathane conformal coating which is used in various places within the optical bench enclosure. This coating releases volatile products, which then deposit onto the optical elements, the cooled detectors, and probably the scan mirror. This material had been used already in GOME 1, where as well a degradation of the optical throughput was observed. The difference to GOME 1 is that in GOME 2, the board surface area has increased by a factor of more than 6, with accordingly more coating amount to act as a source for the off-gassing products. We have modified the description in the revised manuscript as follows:

**Line 44:**

Yet, the volatile coating used within GOME-2's optical bench enclosure makes the optical lens more susceptible to contamination, which eventually leads to instrument degradation (A. Hahne; Munro et al., 2016).

https://doi.org/10.5194/essd-2023-329

Degradation caused by lens contamination may also occur in MODIS VIS/NIR bands, but as far as the analysis we conducted (as shown in Fig 1b), it does not obviously affect MCD43C4 products.

Previous studies have shown wavelength dependence in the degradation caused by coating volatilization. However, the main manifestation is that the degradation is more obvious in the ultraviolet band and weaker in the visible and near-infrared bands. In our manuscript, only band-4 (593–790 nm) of GOME-2 was utilized, in which no obvious wavelength heterogeneity was found in previous studies. Specifically, for the fitting window we used, we added more analysis in the discussion (Sec. 5.1) of the revised manuscript to show the effect of wavelength on the degradation function as below:

**Line 362:**

In addition, the degradation at different wavelengths may also differ. Degradation functions fitted by different wavelengths in the 735–758 nm are compared. A difference of less than 1% was found in the degradation from 2007 to 2021 fitted at different wavelengths (Figure 13a). Similar conclusions can be drawn from Figure 13b. In Figure 13b, it is shown that inconsistency mainly occurs at the Fraunhofer line, which is inherently unstable in time. On the other hand, SIF retrieval relies on the filling of absorption lines. Extremely high fitting accuracy must be ensured if wavelengths are considered an influencing factor of the degradation function. Otherwise, the accuracy of SIF retrieval will be greatly affected. Therefore, in this study, the wavelength dependence of the degradation within the 735–758 nm window is ignored.

[Figure]

Figure 13 (a) The degradation function fitted using reflectance at different wavelengths in 735–758 nm fitting window. The red line is the result obtained at 758 nm, while the degradation functions fitted by other wavelengths were shown in gray. (b) Normalized NIR reflectance spectra in the 735–758 nm fitting window for different years in 2007–2021.

https://doi.org/10.5194/essd-2023-329

The results show that the inconsistency mainly occurs in the Fraunhofer lines, and the maximum fluctuation is 1%. On the other hand, fitting a wavelength-dependent function can result in huge uncertainties, since SIF retrieval is mainly based on the filling effect of SIF to the radiance/reflectance spectra. Slight inaccuracy in the fitting function can change the original depth of the absorption line, which will bring huge errors to SIF retrievals. Therefore, the slight difference in the degradation fitted by different wavelengths was neglected in this manuscript.

**Major comment 2:** A more systematic result session (data validation) is required. The pseudo-invariant method is actually a 2-stage correction to SIF: radiance correction and SIF retrieval. Thus, the method needs to be validated at both stages: comparison of radiance/reflectance to other products such as MODIS, and comparison to other products such as OCO-2 and TROPOMI. The data validation of the radiance/reflectance was missing from the present study.

**Response:** Thanks for this comment. The 2-stage validation was applied to TCSIF in the revised manuscript.

For the validation of GOME-2's radiance, MODIS may not be the most appropriate validation data source, since there are discrepancies in the overpass time, wavelength, and spatial resolution (as shown in Table R1). Measurements of GOME-2C instead of MODIS were utilized in the revised manuscript. A comparison of MODIS, GOME-2A, and GOME-2C was conducted in Table R1 to show the rationality. GOME-2C was launched in November 2018 onboard MetOp-C, which has a similar orbit altitude and spatial resolution as GOME-2A. The parameters of GOME-2A and GOME-2C are highly consistent, most importantly, with the same specification of the NIR band and the same spectral resolution.

**Table R 1. Instrument parameters of GOME-2A, GOME-2C and MODIS**

| Sensor | Overpass time | NIR band (nm) | Ground pixel resolution |
| --- | --- | --- | --- |
| GOME-2A | 9:30 | 590–790 | 80 km × 40 km/ 40 km × 40 km |
| GOME-2C | 9:30 | 590–790 | 80 km × 40 km |
| MODIS | 10:30 / 13:30 | 841~876 | 0.05°×0.05°(500m×500m) |

At the same time, the degradation of GOME-2C can be neglected if we select the measurements in the early stage of the launch. Therefore, the validation of GOME-2A's radiance was conducted in January 2019 using measurements from GOME-C in the revised manuscript.

The introduction of GOME-2C was added in the Datasets part in Sec. 2.2 as follows:

**Line 96:**

**2.2. Datasets for evaluation and comparison**

https://doi.org/10.5194/essd-2023-329

[revised manuscript text omitted]

The analysis of the TROPOMI time series is not shown because the TROPOMI gridded SIF dataset are available only between 2018.4 and 2021.3, which means that there are only two full years of data in 2019 and 2020. Therefore, TROPOMI SIF is currently not able to provide accurate temporal trends.

**Major comment 3:** Following my comment 1, the GOME-2A TOA reflectance at the calibration site (before calibration) has a very clear seasonality in it. Similar seasonality is also found in MODIS data as shown in Fig. 2. I believe the seasonality is due to the BRF effect caused by sun-sensor geometry. But the variation of GOME-2A reflectance seems to be higher than MODIS, why? Also, the GOME-2A reflectance

https://doi.org/10.5194/essd-2023-329

variation seems to increase with time (Fig. 3a). If the degradation is due to the dirty lens, the radiance/reflectance variation should also scale with the Dfactor, right?

**Response:** Thanks for this comment. Quantify analyses were made to show the annual standard deviation and annual range (maximum minus minimum) in the NIR reflectance of GOME-2A and MODIS. The results are shown below:

[Figure]

**Figure R 1 Annual variation of GOME-2A and MODIS NIR reflectance at the calibration site. (a) The annual standard deviation of NIR reflectance. (b) The annual range of NIR reflectance calculated by the annual maximum minus the annual minimum. Blue and orange dots stand for GOME-2A and MODIS, respectively.**

Firstly, the reason for the higher variation of GOME-2A than MODIS may be caused by the effect of the atmosphere. It should be noted that MCD43C4 provides the corrected surface reflectance with atmospheric correction while GOME-2A does not. Although we have reduced the influence of the atmosphere on the TOA reflectance by selecting only pixels with cloud fraction equal to zero, the results may be affected by atmospheric scattering.

Secondly, growing variations were observed in the NIR reflectance of GOME-2A. A similar result was found in GOME-2A chlorine dioxide products (Pinardi et al., 2022), while the authors speculated that the increased noise in GOME-2A after 2011 came from the degradation of the instrument. The results suggested that the instrument degradation may affect the radiance spectra in other forms apart from the contamination of the lens. However, the pattern of such an effect is not easy to find, since the trend is slight (with an average annual change less than 0.05%) and not significant (p-value>0.1).

Compared with the inevitable intra-annual variations, the fitting of the general interannual trend is the main focus of our study. We have added relative analysis to the discussion as follows:

==Line 391:==

https://doi.org/10.5194/essd-2023-329

Besides, the contamination of the lens may not be the only reason for GOME-2A's degradation. As shown in Figure 3, the annual variations in NIR reflectance do not decrease as the annual average does. On the contrary, the fluctuation is growing with time. A similar phenomenon was found in the chlorine dioxide products (Pinardi et al., 2022) that GOME-2A results are noisier than those of GOME-2B, especially after 2011. These results suggest that in addition to the decline in reflectance over time caused by lens contamination, the temporal degradation is impacting GOME-2A measurements in other forms. However, the pattern of this effect is not clear now, further research is needed on the impact of GOME-2A's degradation on its measurements in more aspects. Therefore, only the interannual decline trend was considered in this study, while the inevitable intra-annual variations caused by other factors such as BRF and atmosphere were neglected.

**Minor comment 1:** Line 10. SIF cannot provide a "direct way" to monitor photosynthesis. SIF and GPP are not linearly correlated.

**Response:** Thanks for this comment. The sentence has been revised as follows:

Satellite-based solar-induced chlorophyll fluorescence (SIF) **serves as a valuable proxy for** monitoring the photosynthesis of vegetation globally.

**Minor comment 2:** Line 11. I thought the TROPOMI and OCO SIF datasets were more popular. It is not necessary to say it is the most popular.

**Response:** Thanks for this comment. The SIF products of TROPOMI and OCO-2 are indeed popular because of their high spatial resolution. At the same time, GOME-2 is currently the sensor that can provide the longest SIF time series. From this perspective, we believe that GOME-2 SIF would be of great significance without the problem of time decay. We modified the text to:

Global Ozone Monitoring Experiment-2A (GOME-2A) SIF product **has gained widespread popularity, particularly** due to its extensive global coverage since 2007.

**Minor comment 3:** Line 28. of monitoring -> to proxy.

**Response: Thanks, the sentence has been revised as follows:**

Solar-induced chlorophyll fluorescence (SIF) retrieved from satellite-based hyperspectral data provides a new way **to proxy** the photosynthesis of vegetation globally.

**Minor comment 4:** Line 45. The contamination assumption does not seem to be able to explain the variations in TOA reflectance

https://doi.org/10.5194/essd-2023-329

**Response:** Thanks for this comment. Previous research has attributed sensor degradation to lens contamination with volatile coatings (A. Hahne; Munro et al., 2016). The sentence in Line 45 has been revised as follows to show the details:

Yet, the volatile coating used within GOME-2's optical bench enclosure makes the optical lens more susceptible to contamination, which eventually leads to instrument degradation (A. Hahne; Munro et al., 2016).

As analyzed in major comment 3, BRF, atmospheric scattering, as well as other forms of sensor degradation are affecting the intra-annual variation of GOME-2A's spectra. However, such effects are irregular and have little effect on fitting the general trend (as shown in major comment 3). Therefore, these inevitable variations are ignored in the manuscript, and the limitation was added in the discussion as shown in major comment 3.

**Minor comment 5:** Line 91-95. The use of external PAR to rescale SIF makes the SIF data more prone to errors in external data. Also, since the GOME-2A radiance is corrected using external NIR data from MODIS, should it be considered an L3 product? My understanding is that the L2 product is purely inferred from the L1 product without any external data correction.

**Response:** Thanks for this comment. It is inevitable to introduce external signals (such as PAR, EVI, or fPAR, as shown by Eq.(10) in the manuscript.) while extending the instantaneous SIF signal to the daily average. PAR instead of cos(SZA) was used in the temporal extend model, which indeed adds an external variable. However, as shown in the manuscript, using PAR is a better solution to compensate for the significant effects of diurnal weather changes due to cloud and atmospheric scattering. The results using PAR are shown to be better in the research of Hu et al. (2018). In addition, the instantaneous SIF and daily SIF were provided simultaneously in the Level 2 product. Using instantaneous SIF will avoid the impact of external data.

Also, there are several reasons for defining the un-gridded TCSIF products as Level 2 products:

Firstly, the MCD43C4 products were only used to prove the reliability of the pseudo-invariant site, and not directly used for the correction. The degradation function was fitted merely based on GOME-2A NIR reflectance at the pseudo-invariant site.

Secondly, such a definition makes it easier to distinguish the un-gridded data and the gridded data and keep consistent with other satellite SIF products such as NASA SIF.

Therefore, the un-gridded TCSIF is still defined as Level 2 products in the revised manuscript.

**Minor comment 6:** Line 96. Should it be better to use MODIS data to validate the corrected GOME-2A

https://doi.org/10.5194/essd-2023-329

radiance/reflectance and other SIF data to validate the SIF product? The NDVI/GPP can only be used as indirectly supporting results.

**Response:** Thanks for this comment, we agree that NDVI and GPP can only serve as comparable parameters, rather than benchmarks for validations. We have moved these comparisons to discussion and added a more direct comparison in the results with OCO-2 SIF, and TROPOMI SIF following the previous suggestions, **as detailed in major comment 2**.

It is also necessary to validate the accuracy of the temporally corrected radiance before SIF retrieval. However, MODIS will not be an ideal choice for the validation since there are differences in the spatial resolution, spectral band, and spectral resolution between MODIS and GOME-2A. Instead, we find that GOME-2C measurements can serve as a benchmark. Please find the detailed analysis in the major comment 2.

**Minor comment 7:** Fig. 1. How about other bands? Are they also very stable?

**Response:** Thanks for this comment. In Figure R 2, we show the MCD43C4 Red reflectance (Band 1) as a comparison for the NIR Band. Reflectance at the red band shows similar yearly cycles and interannual variations with NIR reflectance.

[Figure]

**Figure R 2 The surface reflectance at the red and NIR band of the MCD43C4 product.**

Yes, the other bands of MODIS also show stable interannual trends. Since we only use 735–758 nm wavelengths for SIF retrieval in this study, only the NIR band of MCD43C4 was manifested in the manuscript.

**Minor comment 8:** Fig. 2. Since there is no vegetation in the calibration site, the spectral curves should align to the same standard curve when rescaled properly? If you rescale them, is it the case? If so, the Dfactor method is fine; if not, the Dfactor needs to be a wavelength-dependent function!

https://doi.org/10.5194/essd-2023-329

**Response:** Thanks for this advice, we have normalized the radiance spectra by the difference normalization method, the results are added in Figure 13 of the revised manuscript (**as detailed in Major comment 1**). The results show little discrepancies between different wavelengths in our selected fitting window for SIF retrieval (735–758 nm). The main difference appears at the Fraunhofer absorption line between 745 and 758 nm, while the normalized reflectances almost overlap at other wavelengths. Detailed explanations for not considering wavelength in the degradation function were given in major comment 1. Firstly, the deviation caused by wavelength is neglectable. Secondly, to ensure the accuracy of the SIF retrieval, it is better not to include wavelength as an additional independent variable to fit the degradation function without understanding the principle of change.

**Minor comment 9:** Line 221. Is the EVI used as a f_APAR here?

Response: Thanks for this comment. Yes, the EVI is used as a substitution for f_APAR here. Previous studies (Hu et al., 2021) have demonstrated that the FAPAR can be directly quantified by spectral vegetation indices such as NDVI and EVI, and can be estimated using a VI-based linear stretching model (Sellers et al., 1994; Jiang et al., 2002; Liu et al., 2017, 2020). Therefore, the retrieved parameter f_APAR was replaced with EVI as a directly observable parameter.

**Minor comment 10:** Fig. 3. The degradation looks to be an exponential curve, why use a polynomial function to fit it?

**Response:** Thanks for this comment. We have conducted quantified experiments, and the results show that the polynomial function can better fit the data (with higher $R^2$, as shown in Figure R 3). Meanwhile, we select the polynomial function in this study because it is simple to provide robust estimation.

[Figure]

**Figure R 3 Degradation function fitted for GOME-2A NIR reflectance using exponential and**

https://doi.org/10.5194/essd-2023-329

**polynomial models**

**Minor comment 11:** Fig. 6. Combine it with Fig. 5.

**Response:** Thanks for this comment. The two figures have been combined into the new Figure 6 in the revised manuscript as follows:

Line 292:

[Figure]

**Figure 6. Global patterns in the upscaled monthly TCSIF(a, b) and standard error of the weighted mean ($\sigma(F_s)$)(c, d) in July (a, c) and December (b, d) in the year 2008.**

**Minor comment 12:** Fig. 7. You also need to show examples of how the TCSIF and NASA SIF differ in the year 2021 (or more recently) to show the degradation effects.

**Response:** Thanks for this comment. Comparisons between NASA SIF and TCSIF in recent years are better to show the effects of degradation. However, NASA SIF did not provide data beyond 2019, thus, the comparisons in July 2017 and January 2018 were conducted instead. The revision is as below:

Line 286:

We also compared the spatially matched TCSIF and NASA SIF pixels **in January and July 2008, July 2017, and January 2018** (Figure 7a–d). The linear relationships between the two SIF products revealed these to be strongly correlated ($R^2 > 0.65$), significant ($p$-value $< 0.05$), and close to the 1:1 correspondence line

https://doi.org/10.5194/essd-2023-329

(slope > 0.84) for either season in 2008 (Figure 7a, b). **For comparison in 2017 and 2018 (Figure 7c, d), there are still good linear relationships between the TCSIF and NASA SIF ($R^2$ > 0.64). However, it is worth noticing that the regression line deviates from the 1:1 line in both 2017 and 2018(slope < 0.80), which was caused by the degradation in NASA SIF.**

Line 300:

[Figure]

Figure 7. Comparison of TCSIF vs. NASA SIF on 14 January (a) and 15 July (b) in the year 2008, **15 July 2017 (c), and 14 January 2018 (d).** Comparison of TCSIF vs. OCO-2 in July 2019(e) and TCSIF vs. TROPOMI SIF in July 2021 (f). The comparison was made based on the level 2 product. Co-located pixels over land with a cloud fraction < 0.3 have been selected. The color of the scatter points represents the density of the points. The blue dotted line and the black solid line represent the line fitted based on the scatter points and the 1:1 line, respectively.

https://doi.org/10.5194/essd-2023-329

It can already be seen that the discrepancy between NASA SIF and TCSIF increases with time in the results. To compensate for the lack of NASA SIF data, we have also added comparisons between TCSIF, OCO-2 SIF, and TROPOMI SIF in July 2019 and July 2021 to validate the latest results. **As detailed in major comment 2.**

**Minor comment 13:** Line 307. Consider moving these indirect results to a separate section to the very end or discussion.

**Response:** Thanks for this advice, we have moved the indirect comparison with GPP and NDVI to the discussion of the revised manuscript (Sec. 5.4), the comparison with other SIF products is moved to Sec.5.3. Instead, validation with GOME-2C, OCO-2, and TROPOMI was added in the result.

More direct validation of SIF to OCO/TROPOMI and reflectance to MODIS are required.

Thanks again. Please see the response for **major comment 2** for further validation.

Hahne, A.: Investigation on GOME-2 throughput degradation, 2012.

Holdak, A., Kokhanovsky, A., Livschitz, J., and Eisinger, M.: The GOME-2 instrument on the Metop series of satellites: instrument design, calibration, and level 1 data processing – an overview, Atmos. Meas. Tech., 9, 1279–1301, https://doi.org/10.5194/amt-9-1279-2016, 2016.

Hu, J., Liu, L., Guo, J., Du, S., and Liu, X.: Upscaling Solar-Induced Chlorophyll Fluorescence from an Instantaneous to Daily Scale Gives an Improved Estimation of the Gross Primary Productivity, Remote Sensing, 10, 1663, 2018.

Hu, Jiaochan, et al. "Upscaling GOME-2 SIF from clear-sky instantaneous observations to all-sky sums leading to an improved SIF–GPP correlation." Agricultural and Forest Meteorology 306 (2021): 108439.

Jiang, D., et al. "Dynamic properties of absorbed photosynthetic active radiation and its relation to crop yield." Syst. Sci. Compr. Stud. Agric 18 (2002): 51-54.

Liu, Liangyun, Linlin Guan, and Xinjie Liu. "Directly estimating diurnal changes in GPP for C3 and C4 crops using far-red sun-induced chlorophyll fluorescence." Agricultural and Forest Meteorology 232 (2017): 1-9.

Liu, L., Liu, X., Chen, J., Du, S., Ma, Y., Qian, X., ... & Peng, D. (2020). Estimating maize GPP using near-infrared radiance of vegetation. Science of Remote Sensing, 2, 100009.

Munro, R., Lang, R., Klaes, D., Poli, G., Retscher, C., Lindstrot, R., Huckle, R., Lacan, A., Grzegorski, M., Holdak, A., Kokhanovsky, A., Livschitz, J., and Eisinger, M.: The GOME-2 instrument on the Metop

https://doi.org/10.5194/essd-2023-329

series of satellites: instrument design, calibration, and level 1 data processing – an overview, Atmos. Meas. Tech., 9, 1279-1301, https://doi.org/10.5194/amt-9-1279-2016, 2016.

Pinardi, G., Van Roozendael, M., Hendrick, F., Richter, A., Valks, P., Alwarda, R., Bognar, K., Frieß, U., Granville, J., Gu, M., Johnston, P., Prados-Roman, C., Querel, R., Strong, K., Wagner, T., Wittrock, F., and Yela Gonzalez, M.: Ground-based validation of the MetOp-A and MetOp-B GOME-2 OClO measurements, Atmos. Meas. Tech., 15, 3439-3463, 10.5194/amt-15-3439-2022, 2022.

Sellers, P. J., Tucker, C. J., Collatz, G. J., Los, S. O., Justice, C. O., Dazlich, D. A., & Randall, D. A. (1994). A global 1 by 1 NDVI data set for climate studies. Part 2: The generation of global fields of terrestrial biophysical parameters from the NDVI. International Journal of remote sensing, 15(17), 3519-3545.

---

## Author Response (AR3)

https://doi.org/10.5194/essd-2023-329

**Response to reviewers and changes in the revised manuscript**

Dear topic editor and reviewers:

Please find below the remarks from the reviewers (in black), followed by our responses (in blue) and the revised portion of the manuscript (in purple).

Apart from correcting according to the comments of the reviewers, we also corrected a technical error (Line 444: LTSIF_c* -> LT_SIFc*) and added acknowledgments for help with data sources in the Acknowledgments.

Thanks a lot for your careful review.

Chu Zou

zouchu20@mails.ucas.ac.cn
Aerospace Information Research Institute, Chinese Academy of Sciences
No.9 Dengzhuang South Road, Haidian District, Beijing 100094

https://doi.org/10.5194/essd-2023-329

**Response to Reviewer 1 Comments:**

I really appreciate the authors' very detailed responses to my previous comments. I have some follow-up comments that the authors may find helpful:

Thank you for all the suggestions, they are indeed improving this manuscript. Please find our point-to-point reply below.

**Line 20:** I understand that "weather conditions" refers to cloud conditions and atmospheric scattering, not just PAR. I'd suggest specifying them here, otherwise readers may consider some other meteorological variables (e.g., temperature, humidity).

**Response:** Thanks, we have specified that as below:

Besides, a photosynthetically active radiation (PAR)-based upscaling model was employed to upscale the instantaneous clear-sky observations to monthly average values to compensate for the changes in **cloud conditions and atmospheric scattering**.

**Data availability:** while the authors added the download links for a number of SIF datasets, the access links for other datasets are still missing (e.g., GOME-2A/2C radiance and MERRA-2/MODIS datasets). I think such information is critical especially for a dataset paper. Maybe consider adding a summary in the "Data availability" section or a table in the supplementary materials.

**Response:** Thanks. We have added a table showing all the datasets in the supplementary material as below.

**Appendix A. Supplementary material**

**Table S1. Access to the dataset used to generate and compare TCSIF products.**

| Dataset Name | Description | Access |
|---|---|---|
| GOME-2A/C Radiance | Level-1B product of GOME-2A and GOME-2C. | https://data.eumetsat.int/data/map/EO:EUM:DAT:METOP:GOMEL1 |
| Merra-2 PAR | Merra-2 meteorological assimilation reanalysis data (photosynthetically active radiation). | https://goldsmr4.gesdisc.eosdis.nasa.gov/data/MERRA2/M2T1NXRAD.5.12.4/ |
| MODIS MOD13C1 | MODIS Vegetation Indices 16-Day (Version 6.1). | https://lpdaac.usgs.gov/products/mod13c1v061/ |
| MODIS MOD43C4 | The MODIS Version 6.1 Nadir Bidirectional reflectance distribution Adjusted Reflectance (NBAR) product. | https://lpdaac.usgs.gov/products/mcd43c4v006/ |
| LT_SIFc* | Temporally corrected, global 0.05° monthly SIF product. | https://doi.org/10.6084/m9.figshare.21546066.v1 |

| SIFTER | Level-2 daily GOME-2A SIF product accounts for temporal degradation. | https://www.temis.nl/surface/sif.php |
|---|---|---|
| NASA SIF | Level-2 daily SIF (at 740 nm) dataset from GOME-2A. | https://daac.ornl.gov/SIF-ESDR/guides/MetOpA_GOME2_SIF |
| GOSIF | Global 0.05° monthly product of SIF derived from OCO-2, MODIS, and reanalysis data. | https://globalecology.unh.edu/data/GOSIF.html |
| OCO-2 SIF | Level-2 daily SIF (at 740 nm) dataset from OCO-2. | https://disc.gsfc.nasa.gov/datasets/OCO2_L2_Lite_SIF_10r/summary |
| TROPOMI SIF | Level-2 daily SIF (at 740 nm) dataset from TROPOMI. | ftp://fluo.gps.caltech.edu/data/tropomi/ |
| Trendy GPP | Global 0.5° monthly GPP based on the Dynamic Global Vegetation Model. | https://blogs.exeter.ac.uk/trendy/ |
| Pmodel GPP | Global 0.5° daily GPP based on a LUE model (P-model). | https://zenodo.org/records/1423484 |
| MODIS GPP | 8-day composite, 500 m GPP product product based on the radiation use efficiency concept. | https://lpdaac.usgs.gov/products/mod17a2hv061/ |

**Line 97-98:** This sentence should be revised accordingly, regarding the addition of GOME-2C radiance. Maybe consider introducing a bit more about the two-step framework (one for radiance, one for SIF).

**Response:** Thanks. We have modified the description as below:

The dataset was verified through a two-step verification, i.e., the verification of the corrected radiance (compared to radiance measurements in the absence of sensor degradation) and SIF retrievals (compared to other long-term products).

**Fig 3b:** I appreciate the authors' explanation about the "normalized coefficients". However, what is its difference with Dfactor? If they're the same, I'd suggest using the same term across the manuscript (e.g., also in text and Fig. 13a) to avoid confusion.

**Response:** Thanks for the advice. In the last version of this manuscript, we defined the original data in Fig. 3b as "normalized coefficient", and the quadratic function used for temporal correction as "Dfactor". But their physical meanings are indeed the same. The term was changed to "Degradation Factor (Dfactor)" in the figures and text in the revised manuscript.

**Line 264:** GOME-C -> GOME-2C

**Response:** Thanks. It has been corrected.

https://doi.org/10.5194/essd-2023-329

**Line 349:** relative -> related?

**Response:** Thanks, that was a mistake and has been corrected.

**Line 361:** it is interesting that the authors chose to plot the bar as "twice the yearly standard deviation". I am curious why the authors chose twice the standard deviation instead of one (which seems more commonly used?).

**Response:** Thanks. The bar chart shows the average ± one standard deviation, so the length of the bar here becomes twice the standard deviation. We have made it more cleaer in the revised version as below:

[Figure]

Figure 12. Instrument degradation at four different calibration sites. **Each bar shows the yearly average ± standard deviation.**

**Line 364:** I am confused (1) What information does Fig. 13b convey? (2) Why can we draw a similar conclusion from Fig. 13a and Fig. 13b?

**Response:** Thanks. As you mentioned in the last round of comments, the decay factor can change with both time and wavelength. A similar conclusion that can be drawn from both subfigures is that the degradation trend does not vary greatly with wavelength (less than 1%). Unlike Figure 13a, Figure 13b shows how temporal decay varies at different wavelengths. Without the affect of wavelength, the spectra in different year should be overlapped after normalization.However, Figure 13b illustrates that inconsistency mainly occurs at the Fraunhofer line. We then explained that for this reason, we cannot do wavelength-based correction at the expense of incorrectly correcting the Fraunhofer lines, as this would have a large impact on SIF retrieval. We have changed the clarification as below to make it clear:

In addition, the degradation at different wavelengths may also differ. Degradation functions fitted by different wavelengths in the 735–758 nm are compared. A difference of less than 1% was found in the degradation from 2007 to 2021 fitted at different wavelengths (**Figure 13a and Figure 13b**). **Figure 13b shows the**

https://doi.org/10.5194/essd-2023-329

**variation of temporal decay at different wavelengths, indicating that inconsistency mainly occurs at the Fraunhofer line**, which is inherently unstable in time. On the other hand, SIF retrieval relies on the filling of absorption lines. Extremely high fitting accuracy must be ensured if wavelengths are considered an influencing factor of the degradation function. Otherwise, the accuracy of SIF retrieval will be greatly affected. Therefore, in this study, the wavelength dependence of the degradation within the 735–758 nm window is ignored.

**Line 375-376:** While the authors stated "the degradation may differ across different radiance levels", I think a plot showing how the correction performs for different radiance levels would be helpful to gauge the correction uncertainty. Fig. 14 only showing the radiance range is probably not sufficient to demonstrate the validity of this framework across different radiance levels.

**Response:** Thanks for this comment, we have added correction uncertainty analysis for different radiance levels as below:

Line 384:

The relative residuals of the corrected GOME-2A NIR radiance on vegetated targets under different radiance levels were analyzed. As shown in Figure 15, the relative residuals are less than 20% when the NIR radiation is greater than 25 mW m$^{-2}$ sr$^{-1}$ nm$^{-1}$, and the averages of the relative residuals are less than 7%. The results indicate that the correction is basically accurate at different radiance levels. However, when the radiance is lower than 25 mW m$^{-2}$ sr$^{-1}$ nm$^{-1}$, the relative residual error reaches 40%. One reason for the result is that low radiance signals are greatly affected by random noise, resulting in poor comparability of GOME-2A and GOME-2C. Besides, the extremely low radiance level cannot be estimated by the correction based on desert pixels. Therefore, the correction results can be inaccurate at pixels with low vegetation coverage or stressed vegetation.

[Figure]

**Figure 15. Relative residual of NIR radiance (calculated as the absolute difference between GOME-2A and GOME-2C NIR radiance at the co-located points) at different radiance levels. Global vegetation targets with SIF signals greater than 0.1 mW m$^{-2}$ sr$^{-1}$ nm$^{-1}$ on July 1, 2019 were selected.**

**Line 392:** fluctuation->degradation?

**Response:** Here by "fluctuation", we mean the variation within every single year. In Figure 3, we found the yearly variation slightly increased with time since 2011, which may be caused by other effects of GOME-2A's contamination. In the revised manuscript, we distinguished the "intra-annual variation" and "inter-annual variation" to avoid confusion.

**Line 404:**

Besides, the contamination of the lens may not be the only reason for GOME-2A's degradation. As shown in Figure 3, the **intra-annual variation** in NIR reflectance does not decrease as the **inter-annual average** does. Instead, the **intra-annual variation** is growing with time. A similar phenomenon was found in the chlorine dioxide products (Pinardi et al., 2022) that GOME-2A results are noisier than those of GOME-2B, especially after 2011.

https://doi.org/10.5194/essd-2023-329

**Response to Reviewer 2 Comments:**

The authors did a great job revising the manuscript and addressed all my comments. My last 3 minor comments are:

1. Line 259: "which results in less atmospheric absorption in GOME-2C", Why less atmosphere absorption? There is (almost?) no air in space, so the satellite's location does not impact the atmospheric absorption (nadir). If the satellite is not directly above, a cos(VZA) would need to be accounted for.

**Response 1:** Thanks for this comment. The reasons mentioned in the previous revision cannot be the reason why the radiance of GOME-2C is slightly lower than that of the corrected GOME-2A radiance. We have deleted this inaccurate description in the revised manuscript. GOME-2A is not a nadir observation, but we only selected the data with VZA less than 20° (we have added related descriptions in the text and figure note in the revised version) and divided the data by cos(VZA), but there are still small systematic errors. In addition to the error in the correction of GOME-2A degradation, we attribute the possible cause of the systematic deviation to the anisotropy of the surface. The text is modified as follows:

**Line 254:**

The temporally corrected GOME-2A NIR radiance was validated using GOME-2C radiance spectra (Figure 4). For the corrected GOME-2A radiance, the scatter plot shows that the majority of points are concentrated near the 1:1 line (Figure 4a). The difference between the two products followed a Gaussian distribution with a small mean value of 1.85 mW $m^{-2}$ $sr^{-1}$ $nm^{-1}$, which is 2.3% of the mean GOME-2A radiance (Figure 4b). On the contrary, the mean deviation without temporal correction is 15.16W $m^{-2}$ $sr^{-1}$ $nm^{-1}$ (Figure 4d). **Slight positive offsets can be found in both linear regression results. The difference in orbit height between GOME-2A (827 km) and GOME-2C (817 km) leads to the difference in viewing zenith angle (VZA). Although only observations with VZA<20° were selected, and the effect of observing angle has been corrected by dividing the cosine of VZA, there may still be differences due to the anisotropy of the ground surface, which introduces systematic errors.**

2. The polynomial fitting would not be an ideal correction, because if you have new data, you will need to redo the fitting, and the data for previous years would need to be updated using the new fitting results.

**Response 2:** Thank you for your comment. It's correct that the fitting function is dependent on the data participating in the fitting. However, the calibration procedure in this study uses the screened data within all available time ranges of GOME-2A (2007.01-2021.11). There won't be any "new data" beyond the time range, so the function is representative of the radiance degradation between 2007 and 2021. The function fits the degradation with high precision ($R^2$=0.851). Thus, we considered the function precise and stable enough to correct the general trend of recession.

https://doi.org/10.5194/essd-2023-329

3. Make corrections to other wavelength rad would be great too, and it will make the data more valuable for users who do not use SIF.

**Response 3:** Thanks for this comment. It's true that the degradation problem also exists in other wavelength bands, and the correction is needed for the use of GOME-2 radiance. However, this is not the focus of this study, because for SIF retrieval, we only utilized NIR band of GOME-2A. Further works on the correction are expected in subsequent works.